# UPrompt: Bidirectional Multi-granularity Learning for Vision-Language Models

## Abstract

The prompt learning paradigm for vision-language models is effective yet faces the dilemma of balancing granularity: global prompts lack fine-grained semantic awareness, while local prompts ignore overall contextual associations, leading to limited cross task generalization. This dilemma exists in dense prediction tasks. Inspired by the U-Net framework that unifying multi-level representations across different granularities, we propose UPrompt, a novel bidirectional multi-granularity prompt learning framework for vision-language models. Similar to how U-Net integrates fine and coarse features through symmetric encoder-decoder pathways with cross-level connections, UPrompt constructs parallel multi-granularity representations in both visual and textual modalities, where coarse-to-fine cascaded enhancement propagates global contextual information to refine local details, while fine-to-coarse hierarchical supervision ensures semantic consistency across scales. Extensive experiments on 17 benchmarks validate our effectiveness. Our method outperforms MAMET and VPKE by +4.1 and +7.3 rSum on MSCOCO, surpasses CoCoA-Mix by +5.09% in base-to-novel generalization, while maintaining competitive performance with minimal overhead (coarse-grained) and matching PSRC with 1/3 cost (medium-grained).

## 1 Introduction

Prompt learning has emerged as an effective paradigm for adapting Vision-Language Models (VLMs) by optimizing learnable prompt tokens Zhou et al. (2022); Khattak et al. (2023). However, a critical limitation arises: existing methods Yao et al. (2023); Roy & Etemad (2024) mainly optimize at a single, fixed granularity, leading to an inherent trade-off between capturing broad contextual information and preserving fine-grained visual details. Global prompting strategies cannot encode localized features for fine-grained reasoning, as CoOp Zhou et al. (2022) with single global prompts underperforms fine-grained TAP Ding et al. by 11.31% on the fine-grained FGVCAircraft dataset, while finely-structured prompts struggle to integrate sufficient global context or model compositional relationships across image regions, as illustrated in Fig. 1. This granularity bottleneck significantly constrains adaptation performance and generalization across diverse vision-language tasks.

In view of this problem, we turn to hierarchical architectures that have demonstrated remarkable success in dense prediction tasks, i.e., the U-Net Ronneberger et al. (2015). The U-Net enables effective multi-scale modeling through its symmetric encoder-decoder design and skip connections, which collectively maintain both high-level context and localized details. Yet, transferring these principles into prompt learning is non-trivial. A fundamental paradigm gap exists: while U-Net operates on spatially structured pixel grids Williams et al. (2023), prompt learning functions in an abstract embedding space Huang et al. (2025); Li et al. (2024) where textual prompts lack inherent geometric structure. This raises two core challenges: how to construct meaningful multi-granularity representations that embody semantic hierarchy in both modalities, and how to establish bidirectional information flow between granularities to ensure consistency and complementary learning.

To address these challenges, we propose UPrompt learning, a U-shaped multi-granularity prompt learning framework that introduces a structured hierarchy into vision-language adaptation, as illustrated in Fig. 1(c). Inspired by U-Net's multi-scale feature fusion mechanisms, e.g., skip connection Ronneberger et al. (2015), UPrompt constructs parallel granularity pathways in vision and language, using progressive spatial pooling for images and iterative semantic enrichment for

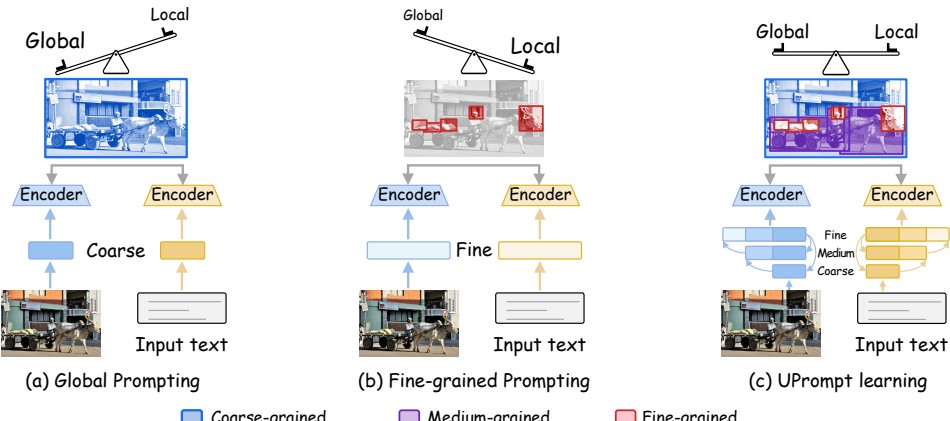

Coarse-grained ☐  Medium-grained ☐  Fine-grained ☐

Figure 1: **The granularity trade-off in prompt learning.** (a) Global prompting captures broad context but lacks fine-grained details, while (b) fine-grained prompting preserves local features but loses global information. (c) UPrompt learning addresses the trade-off through multi-granularity hierarchical modeling, achieving both global understanding and local precision.

text. Moreover, the framework incorporates a bidirectional connection mechanism (analogous to skip-connection in U-Net): a coarse-to-fine cascaded mechanism that injects global context into fine-grained features using cross-granularity attention, and a fine-to-coarse hierarchical supervision that distributes semantic knowledge from the finest to coarsest levels via distillation. This ensures not only enhanced representational capacity at each level, but also consistent semantics across different granularities. Comprehensive evaluation across diverse benchmarks validates the effectiveness of our approach. UPrompt bridges hierarchical representation learning principles from U-Net to prompt-based VLMs, establishing a unified framework that resolves the limitation of single granularity through bidirectional information flow. This architecture provides flexible multi-granularity alignment while ensuring semantic consistency across different representation scales, offering a principled solution for vision-language adaptation. Our main contributions are as follows:

- We introduce UPrompt, a U-Net-inspired framework for prompt learning that leverages hierarchical multi-granularity representations across vision-language modalities to overcome single-scale adaptation limitations.

- We introduce bidirectional connection, establishing bidirectional information flow across multi-granularity hierarchies. Coarse-to-fine enhancement injects global context into fine-grained representations for improved local modeling, while Fine-to-Coarse Supervision leverages finest-level alignment to regularize coarser granularities, ensuring semantic consistency.

- Experiments across 17 benchmark datasets demonstrate UPrompt's superiority in cross-modal retrieval, few-shot classification, base-to-novel generalization, and out-of-distribution scenarios, with our hierarchical design enabling flexible performance-efficiency trade-offs.

## 2 RELATED WORK

**Prompt Learning in VLMs.** Prompt learning, pioneered in NLP, was introduced to VLMs by CoOp Zhou et al. (2022) for CLIP Radford et al. (2021), with extensions to both modalities Khattak et al. (2023); Cho et al. (2023). To overcome single global prompt limitations, subsequent research focuses on diverse multi-granularity representations. GalLoP Lafon et al. (2024) introduces dual prompts for global and local features. TAP Ding et al. achieves diversity via attribute trees. SurPL Liu et al. employs generators for dynamic features. HiCroPL Zheng et al. (2025) builds hierarchical representations through multi-level prompt injection, while SPTR Cui et al. (2025) leverages diverse fixed prompt ensembles. These methods treat different granularities as independent modules combined at final fusion, lacking unified frameworks for cross-scale dependencies and information flow. Our U-Net inspired framework unifiedly enables bidirectional connection for progressive integration and semantic consistency across hierarchical structures.

**Hierarchical Representation.** Fusing multi-scale features is a cornerstone principle for visual understanding, established by FPN Lin et al. (2017b) and U-Net Ronneberger et al. (2015). This

idea has been continuously adapted to new architectures. UNet++ Zhou et al. (2019) improves U-Net by introducing nested skip connections to bridge the semantic gap between encoder and decoder features. GraphFPN Zhao et al. (2021) extends this to graph neural networks by constructing data-dependent feature pyramids. HGFormer Ding et al. (2023) implements part-whole grouping in Vision Transformers for robust representations. MSVMamba Shi et al. (2024) integrates hierarchical design into State Space Models, demonstrating the principle's enduring relevance. VLMs require processing both coarse context and fine details, necessitating corresponding textual representations for each visual granularity. Unlike images, text's unstructured nature prevents simple spatial hierarchical construction. We propose building and aligning multi-granularity hierarchies across modalities.

**Cross-Level Interaction.** Hierarchical representations employ coarse-to-fine and fine-to-coarse paradigms. Coarse-to-fine methods propagate global context to refine local details, as in Stacked Hourglass Networks Newell et al. (2016) and RefineNet Lin et al. (2017a). Fine-to-coarse strategies aggregate low-level features for high-level consistency, exemplified by GroupViT Xu et al. (2022) and HVQ Lu et al. (2023). Advanced architectures like NeRD-Rain Chen et al. (2024) enable bidirectional flow where features are simultaneously refined by coarser context and enriched by finer details. Inspired by such strategies, we address VLM prompt learning's limitation where hierarchical prompts are optimized in isolation, proposing a bidirectional connection framework that enables progressive integration and ensures semantic consistency across the structure.

## 3 METHODOLOGY

### 3.1 PRELIMINARIES

**CLIP.** Contrastive Language-Image Pre-training (CLIP) Radford et al. (2021) uses an image encoder $\mathcal{F}$ and a text encoder $\mathcal{G}$ to map image $x$ and text $t$ into a shared space: $\mathbf{z}_v = \mathrm{norm}(\mathcal{F}(x))$, $\mathbf{z}_t = \mathrm{norm}(\mathcal{G}(t))$. A symmetric contrastive loss aligns matched pairs and separates mismatched ones, enabling zero-shot classification and cross-modal retrieval. With temperature $\tau$, the probability is:

$$p(k|x) = \frac{\exp(\mathrm{sim}(\mathbf{z}_v, \mathbf{z}_{t,k})/\tau)}{\sum_j \exp(\mathrm{sim}(\mathbf{z}_v, \mathbf{z}_{t,j})/\tau)}. \tag{1}$$

**Prompt learning methodology.** Prompt learning efficiently optimizes VLMs by incorporating learnable prompt tokens instead of full fine-tuning approaches Zhou et al. (2022); Khattak et al. (2023). The textual and visual input token sequences at transformer layer $i$ are formally defined as: $T_{input}^{(i)} = \{t_{bos}, P_t^{(i)}, T_{embed}, t_{eos}\}$ and $V_{input}^{(i)} = \{v_{cls}, E_{patch}, P_v^{(i)}\}$, where $P_t^{(i)} = \{p_t^1, p_t^2, ..., p_t^\eta\}$ and $P_v^{(i)} = \{p_v^1, p_v^2, ..., p_v^M\}$ are learnable prompt vectors with dimensions $\mathbb{R}^\eta$ and $\mathbb{R}^M$ respectively.

**U-shaped architecture.** U-shaped networks (e.g., U-Net Ronneberger et al. (2015)) consist of a symmetric encoder-decoder design with skip connections between corresponding layers. Let the network have $L$ levels; the encoder at level $i$ outputs features $\mathbf{h}^{(i)}$, and the decoder at level $i$ fuses them with the upsampled features from level $i + 1$:

$$\tilde{\mathbf{h}}^{(i)} = \phi^{(i)}\big(\mathbf{h}^{(i)}, \mathrm{up}(\tilde{\mathbf{h}}^{(i+1)})\big), \tag{2}$$

where $\phi^{(i)}(\cdot)$ is a cross-level fusion operator and $\mathrm{up}(\cdot)$ denotes upsampling. This enables multi-level information propagation, maintaining global context while preserving fine details.

### 3.2 U-SHAPED MULTI-GRANULARITY PROMPTING

**UPrompt Learning Paradigm.** To address the limitation of trade-off between global context and local details in single-granularity prompt learning, we draw inspiration from U-Net's hierarchical processing where $\tilde{\mathbf{h}}^{(i)} = \phi^{(i)}(\mathbf{h}^{(i)}, \mathrm{up}(\tilde{\mathbf{h}}^{(i+1)}))$ fuses multi-level features, maintaining both global context and fine-grained details across scales. Existing multi-level methods (e.g., TAP, HiCroPL) treat granularities as independent modules or operate within network depths, lacking cross-scale dependencies across semantic hierarchies. We propose U-shaped multi-granularity prompting, dubbed as UPrompt learning, that constructs parallel hierarchical semantic structures with explicit cross-granularity information flow across modalities. We extend CLIP encoders $\mathcal{F}$ and $\mathcal{G}$ to multi-granularity versions $\{\mathcal{F}^{(k)}, \mathcal{G}^{(k)}\}_{k=1}^K$ where $k \in [1, K]$ spans coarsest to finest granularities.

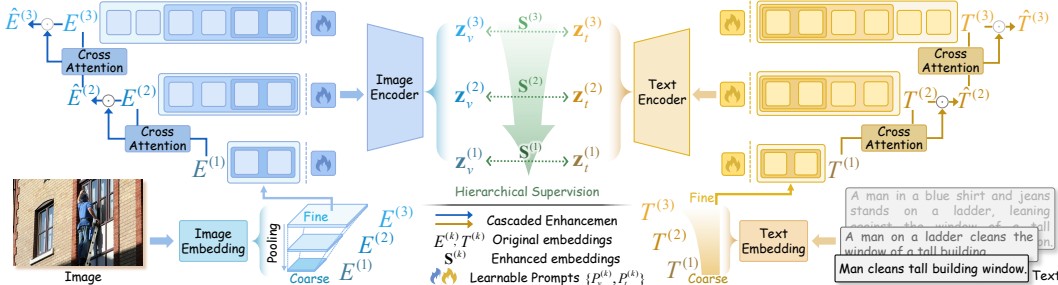

**Figure 2: Method overview of UPrompt.** UPrompt learning constructs hierarchical vision-language alignment via multi-granularity pathways with learnable prompts. Bidirectional connection operates through coarse-to-fine cascaded enhancement that injects global context into fine-grained embeddings via cross-attention, and fine-to-coarse hierarchical supervision that guides coarser levels using finest-grained representations.

For visual modality, we construct nested patch hierarchies through progressive downsampling pooling in embedding space. The input image is first processed by CLIP's patch embedding layer to extract the finest-grained patch tokens $E_{patch}^{(K)}$. Coarser representations are derived via recursive pooling: $E_{patch}^{(k)} = \text{Pool}^{(k)}(E_{patch}^{(k+1)})$ for $k = K-1, \ldots, 1$, ensuring $|E_{patch}^{(k)}| < |E_{patch}^{(k+1)}|$. At each granularity level $k$, we concatenate the pooled patch embeddings with learnable prompts $P_v^{(k)} \in \mathbb{R}^{M \times d}$ and feed them into CLIP's vision encoder to obtain visual features: $\mathbf{z}_v^{(k)} = \mathcal{F}^{(k)}([E_{patch}^{(k)}; P_v^{(k)}])$.

For textual modality, we construct semantic hierarchies via progressive enrichment:

$$T_{embed}^{(1)} = \Phi_{\text{abstract}}(t), \tag{3}$$

$$T_{embed}^{(k)} = T_{embed}^{(k-1)} \oplus \Phi_{\text{refine}}^{(k)}(t), \quad k = 2, \ldots, K, \tag{4}$$

where $\Phi_{\text{abstract}}(\cdot)$ extracts core semantics, $\Phi_{\text{refine}}^{(k)}(\cdot)$ generates granularity-specific elaborations, and $\oplus$ denotes semantic expansion ensuring nesting $T_{embed}^{(k)} \subset T_{embed}^{(k+1)}$. Text representations integrate prompts $P_t^{(k)} \in \mathbb{R}^{\eta \times d}$, i.e., $\mathbf{z}_t^{(k)} = \mathcal{G}^{(k)}([P_t^{(k)}; T_{embed}^{(k)}])$. The operators $\Phi_{\text{abstract}}(\cdot)$ and $\Phi_{\text{refine}}^{(k)}(\cdot)$ are instantiated via LLMs with specific prompts for multi-level text generation (details in Sec. 4).

Cross-modal alignment at each granularity $k$ is achieved through similarity $s^{(k)}(x, t) = \frac{\mathbf{z}_v^{(k)} \cdot \mathbf{z}_t^{(k)}}{\|\mathbf{z}_v^{(k)}\| \|\mathbf{z}_t^{(k)}\|}$. This U-shaped architecture enables multi-granularity vision-language alignment through hierarchical prompt learning across symmetric pathways, with visual branch providing spatial representations and textual branch offering semantic specifications.

### 3.3 BIDIRECTIONAL CONNECTION FOR UPROMPT LEARNING

Simple granularity stacking in UPrompt learning lacks inter-granularity interaction, causing fine-grained context deficiency and coarse-grained optimization inconsistency that limit multi-granularity representation. To address these challenges, we propose bidirectional connection for UPrompt learning, which employs coarse-to-fine cascaded enhancement during forward propagation and fine-to-coarse hierarchical supervision during backward optimization (Fig. 2).

**Coarse-to-Fine Cascaded Enhancement (CE).** To address context deficiency where fine-grained embeddings lack global contextual guidance for modeling local information relationships, we propose cascaded enhancement that injects coarse-grained contextual information into finer embeddings. For embeddings $X^{(k)} \in \{E_{patch}^{(k)}, T_{embed}^{(k)}\}$ at granularity level $k$, the enhancement operation is:

$$\hat{X}^{(k)} = X^{(k)} \odot \mathcal{A}(X^{(k)}, \hat{X}^{(k-1)}), \tag{5}$$

where $\hat{X}$ are enhanced embeddings, $\odot$ is element-wise product, $\mathcal{A}(\cdot, \cdot)$ is cross-granularity attention:

$$\mathcal{A}(X^{(k)}, \hat{X}^{(k-1)}) = \text{softmax}\left(\frac{X^{(k)}\mathbf{W}_q(\hat{X}^{(k-1)}\mathbf{W}_k)^\top}{\sqrt{d}}\right)\hat{X}^{(k-1)}\mathbf{W}_v, \tag{6}$$

where $\mathbf{W}_{q,k,v} \in \mathbb{R}^{d \times d}$ are query, key, and value projection matrices, $d$ is the embedding dimension. Enhanced embeddings are then fed into encoders to obtain $\mathbf{z}_v^{(k)} = \mathcal{F}^{(k)}([\hat{E}_{patch}^{(k)}; P_v^{(k)}])$ and $\mathbf{z}_t^{(k)} = \mathcal{G}^{(k)}([P_t^{(k)}; \hat{T}_{embed}^{(k)}])$. Fine-grained embeddings can thus extract contextually relevant information from global representations, enhancing local information modeling with global contextual guidance.

**Proposition 1** (CE Directional Alignment Effect). *Let $\hat{X}^{(k)}$ be the fine-grained representation at level $k$ enhanced by coarse-to-fine cascaded enhancement (CE, Eq. (5)-(6)), which leverages contextual guidance from the coarser representation $\hat{X}^{(k-1)}$. Let $X^{(k)}$ be its unenhanced counterpart. Under the mild assumption that the coarse context is informative, CE provably strengthens the alignment between fine-grained features and their coarse-grained guidance in expectation:*

$$\mathbb{E}\left[\frac{\langle \hat{X}^{(k)}, \hat{X}^{(k-1)} \rangle}{\|\hat{X}^{(k)}\|\|\hat{X}^{(k-1)}\|}\right] \geq \mathbb{E}\left[\frac{\langle X^{(k)}, \hat{X}^{(k-1)} \rangle}{\|X^{(k)}\|\|\hat{X}^{(k-1)}\|}\right]. \tag{7}$$

*Proof.* Cascaded enhancement injects coarse contextual guidance into fine representations via cross-attention, provably improving directional alignment in expectation. See Appendix A.1.

**Fine-to-Coarse Hierarchical Supervision (HS).** To address optimization inconsistency from semantic drift at coarse granularities, we propose fine-to-coarse hierarchical supervision using the superior alignment of finest-grained features. The finest-level features $(\mathbf{z}_v^{(K)}, \mathbf{z}_t^{(K)})$ achieve optimal cross-modal correspondence through rich representational capacity, serving as teacher signals for coarser levels. The finest-level cross-modal similarity matrix $\mathbf{S}^{(K)}$ provides the teacher distribution:

$$\mathbf{S}_{ij}^{(K)} = \frac{\mathbf{z}_{v,i}^{(K)} \cdot \mathbf{z}_{t,j}^{(K)}}{\|\mathbf{z}_{v,i}^{(K)}\|\|\mathbf{z}_{t,j}^{(K)}\|}. \tag{8}$$

All coarser levels are supervised via knowledge distillation from the detached finest-level representations, preventing degradation of the teaching signal by coarse-grained semantic drift:

$$\mathcal{L}_{\text{guide}} = \frac{1}{K-1} \sum_{k=1}^{K-1} \mathbb{E}_{(i,j)}\left[D_{\text{KL}}\left(\text{softmax}\left(\mathbf{S}_{i,:}^{(k)}/\tau_d\right) \| \text{softmax}\left(\mathbf{S}_{i,:}^{(K)}/\tau_d\right)\right)\right]. \tag{9}$$

where $\tau_d$ is the distillation temperature. Detaching $\mathbf{S}^{(K)}$ prevents gradients from coarse-level training flowing back to the finest layer, ensuring that fine-grained misalignment does not corrupt coarse representations. Concurrently, CE injects global contextual guidance into fine-grained features, enabling self-correction across granularities. This hierarchical supervision enforces semantic consistency across all granularity levels, preventing coarse-grained drift while maintaining the complementary nature of multi-granularity representations that enables more effective cascaded enhancement.

**Proposition 2** (HS Consistency and Substitutability). *Let $S^{(k)}$ and $S^{(K)}$ be similarity matrices from Eq. (8), and define $p_{\tau_d}^{(k)}(j|i) = softmax(S_{i,:}^{(k)}/\tau_d)_j$ and $q_{\tau_d}^{(K)}(j|i) = softmax(S_{i,:}^{(K)}/\tau_d)_j$ where teacher $q^{(K)}$ is detached as in Eq. (9). Assuming HS aligns coarse-grained distributions with fine-grained teachers, HS bounds semantic drift and enables performance-preserving coarse inference:*

$$\mathbb{E}_{(x,t),i}\left[\text{KL}\left(q_{\tau_d}^{(K)}(\cdot|i) \| p_{\tau_d}^{(k)}(\cdot|i)\right)\right] \leq \varepsilon \implies \mathbb{E}_{(x,t),i}\left[|\Phi\left(p_{\tau_d}^{(k)}(\cdot|i)\right) - \Phi\left(q_{\tau_d}^{(K)}(\cdot|i)\right)|\right] \leq L\sqrt{\varepsilon/2}, \tag{10}$$

*for any $L$-Lipschitz functional $\Phi$ w.r.t. total variation distance. The detach operation ensures gradient isolation: $\partial L_{guide}/\partial z^{(K)} = 0$.*

*Proof.* Hierarchical supervision constrains KL divergence between coarse and fine distributions, yielding bounded substitutability via Pinsker's inequality. See Appendix A.2.

**Overall Objective.** The UPrompt learning framework combines contrastive alignment loss across all $K$ granularity levels with hierarchical supervision for cross-modal alignment and inter-granularity consistency. The contrastive losses are averaged for stable optimization:

$$\mathcal{L}_{\text{UPrompt}} = \mathcal{L}_{\text{guide}} + \frac{1}{K} \sum_{k=1}^{K} \mathbb{E}_{(x,t)}\left[-\log \frac{\exp(\text{sim}(\mathbf{z}_v^{(k)}, \mathbf{z}_t^{(k)})/\tau)}{\sum_{t'} \exp(\text{sim}(\mathbf{z}_v^{(k)}, \mathbf{z}_{t'}^{(k)})/\tau)}\right]. \tag{11}$$

Table 1: **Cross-modal retrieval performance** of different CLIP fine-tuning methods. "ZS" denotes zero-shot and "FT" is fine-tuned. rSum is the sum of all R@1, R@5, and R@10 scores. Best results highlighted in first , second .

| Methods | Flickr30K | | | | | | | MSCOCO | | | | | | |
| | Image-to-Text | | | Text-to-Image | | | | Image-to-Text | | | Text-to-Image | | | |
| | R@1 | R@5 | R@10 | R@1 | R@5 | R@10 | rSum | R@1 | R@5 | R@10 | R@1 | R@5 | R@10 | rSum |
|---|---|---|---|---|---|---|---|---|---|---|---|---|---|---|
| CLIP$_{ZS}$ | 81.3 | 96.4 | 98.5 | 62.2 | 85.7 | 91.7 | 515.8 | 52.5 | 76.6 | 84.7 | 33.1 | 58.4 | 69.0 | 374.3 |
| CLIP$_{FT}$ | 91.7 | 99.0 | 99.5 | 79.1 | 95.2 | 97.6 | 562.1 | 66.9 | 88.3 | 93.6 | 51.5 | 78.0 | 86.1 | 464.4 |
| DoPL$_{(ACL'25)}$ | 69.8 | 90.7 | 95.0 | 66.9 | 89.0 | 93.6 | 505.0 | 63.2 | 86.7 | 91.8 | 49.6 | 76.3 | 85.2 | 452.8 |
| MAMET$_{(TCSVT'25)}$ | 92.7 | 99.3 | 99.7 | 79.8 | 95.2 | 97.2 | 563.9 | 66.0 | 88.4 | 93.6 | 52.4 | 79.3 | 87.3 | 467.0 |
| VPKE$_{(TCSVT'25)}$ | 93.7 | 99.2 | 99.8 | 82.0 | 95.7 | 98.2 | 568.6 | 69.2 | 89.0 | 94.2 | 52.8 | 78.5 | 86.5 | 470.2 |
| UPrompt | 93.8 | 99.4 | 99.6 | 83.6 | 96.3 | 98.4 | 571.1 | 70.1 | 89.8 | 94.8 | 52.6 | 79.1 | 87.9 | 474.3 |

During inference, we can flexibly select granularity levels based on performance-efficiency trade-offs. We default to finest-grained features $(\mathbf{z}_v^{(K)}, \mathbf{z}_t^{(K)})$ for optimal performance. When prioritizing efficiency, coarser levels offer reduced token requirements and lower costs while preserving semantic consistency through our fine-to-coarse hierarchical supervision that prevents coarse-grained drift.

## 4 EXPERIMENTS

**Datasets.** For cross-modal retrieval, we evaluate on Flickr30K Young et al. (2014) with 31,783 images and MSCOCO-5K Lin et al. (2014) with 123,287 images, each annotated with 5 captions. For classification tasks, we use 11 datasets: ImageNet Deng et al. (2009), Caltech101 Fei-Fei et al. (2004), OxfordPets Parkhi et al. (2012), StanfordCars Krause et al. (2013), Flowers102 Nilsback & Zisserman (2008), Food101 Bossard et al. (2014), FGVCAircraft Maji et al. (2013), SUN397 Xiao et al. (2016), UCF101 Soomro et al. (2012), DTD Cimpoi et al. (2014) and EuroSAT Helber et al. (2019). For out-of-distribution evaluation, we use ImageNet-A Hendrycks et al. (2021b), ImageNet-R Hendrycks et al. (2021a), ImageNet-Sketch Wang et al. (2019), and ImageNet-V2 Recht et al. (2019).

**Implementation details.** For multi-granularity construction, visual modality applies progressive spatial pooling to patch embeddings: from original 14×14 to 7×7, 4×4, and 1×1 tokens. Classification uses all 4 scales while retrieval uses the first 3 scales. For textual modality, we employ Llama 3-8B to generate hierarchical representations. In classification, we construct 4-level prompts:"*a photo of a {cls}*" (Level 1), progressively enriched with representative nouns (Level 2), attribute phrases (Level 3), and detailed descriptions (Level 4). In retrieval, original captions serve as medium granularity, with coarse granularity created via prompt"*shorten the caption and keep important information*", and fine granularity enhanced via "*add details from other captions to enhance the original caption and keep original meaning unchanged*". Text hierarchies (via LLM) are pre-computed rather than computed in real-time, avoiding significant computational overhead. We use CLIP ViT-B/16 as backbone. We employ 3- and 4-granularity configurations for retrieval and classification respectively, with distinct learnable prompts of length 4 for each granularity level. During inference, we default to the finest-grained features, which consistently yield optimal alignment.

### 4.1 COMPARATIVE RESULTS

**Cross-modal retrieval.** We evaluate on Flickr30K and MSCOCO using Recall@K (K=1,5,10) and rSum (Table 1). UPrompt achieves rSum scores of 571.1 and 474.3 on Flickr30K and MSCOCO, outperforming recent CLIP-based fine-tuning methods. It surpasses DoPL Guo et al. (2025) (505.0, 452.8), which generates prompts for each layer to achieve alignment, and MAMET Wang et al. (2025b) (563.9, 467.0), which distills knowledge from multiple embeddings. UPrompt also shows clear advantage over VPKE Wang et al. (2025a) (568.6, 470.2), which enhances retrieval by using external visual knowledge. UPrompt's superior results, evidenced by R@1 scores of 93.8% on Flickr30K and 70.1% on MSCOCO, stem from its unique architecture. Its multi-granularity prompting captures vision-language alignments across various semantic levels, and our method provides contextual guidance from coarse to fine granularities, leading to significant improvement in retrieval precision.

**Base-to-novel generalization.** UPrompt achieves an 82.12% harmonic mean (HM) across 11 datasets (Table 2), achieving a +1.06% average HM over the second best method. It outperforms recent multi-level methods like TAP (81.04%) Ding et al., which constructs concept-attribute hierarchies,

Table 2: **Base-to-novel generalization.** Bold values indicate the best results. HM: Harmonic Mean.

| Method | Average | | | ImageNet | | | Caltech101 | | | OxfordPets | | |
|---|---|---|---|---|---|---|---|---|---|---|---|---|
| | Base | Novel | HM | Base | Novel | HM | Base | Novel | HM | Base | Novel | HM |
| CoOp(IJCV'22) | 82.69 | 63.22 | 71.66 | 76.47 | 67.88 | 71.92 | 96.00 | 89.81 | 93.73 | 93.67 | 95.29 | 94.47 |
| PSRC(ICCV'23) | 84.26 | 76.10 | 79.97 | 77.60 | 70.73 | 74.01 | 98.10 | 94.03 | 96.02 | 95.33 | 97.30 | 96.30 |
| TAP(ICLR'25) | 84.75 | 77.63 | 81.04 | 77.97 | 70.40 | 73.99 | **98.90** | 95.50 | 97.17 | 95.80 | 97.73 | 96.76 |
| CLIP-AST(CVPR'25) | 85.64 | 76.99 | 81.06 | 78.44 | 70.22 | 74.10 | 98.71 | 94.00 | 96.30 | 96.23 | 97.37 | 96.80 |
| SurPL-G(ICML'25) | **86.37** | 76.32 | 81.03 | **78.74** | 70.49 | 74.39 | 98.77 | 95.16 | 96.93 | 96.37 | 97.41 | 96.89 |
| CoCoA-Mix(ICML'25) | 79.31 | 75.10 | 77.03 | 75.47 | 68.92 | 72.04 | 98.02 | 94.39 | 96.17 | 95.16 | 97.60 | 96.36 |
| UPrompt(Ours) | 86.35 | **78.29** | **82.12** | 78.65 | **71.24** | **74.76** | 98.78 | **95.84** | **97.29** | **96.41** | **97.92** | **97.16** |

| Method | StanfordCars | | | Flowers102 | | | Food101 | | | FGVCAircraft | | |
|---|---|---|---|---|---|---|---|---|---|---|---|---|
| | Base | Novel | HM | Base | Novel | HM | Base | Novel | HM | Base | Novel | HM |
| CoOp(IJCV'22) | 78.12 | 60.40 | 68.13 | 97.60 | 59.67 | 74.06 | 88.33 | 82.26 | 85.19 | 40.44 | 22.30 | 28.75 |
| PSRC(ICCV'23) | 78.27 | 74.97 | 76.58 | 98.07 | 76.50 | 85.95 | 90.67 | 91.53 | 91.10 | 42.73 | 37.87 | 40.15 |
| TAP(ICLR'25) | 80.70 | 74.27 | 77.35 | 97.90 | 75.37 | 85.30 | 90.97 | 91.83 | 91.40 | 40.40 | 36.50 | 40.06 |
| CLIP-AST(CVPR'25) | **84.21** | 74.05 | 78.80 | 97.91 | 77.73 | 86.66 | 90.57 | 91.11 | 90.84 | 48.98 | 38.21 | 42.93 |
| SurPL-G(ICML'25) | 83.57 | 72.77 | 77.80 | **98.90** | 72.88 | 83.92 | 90.92 | 91.81 | 91.36 | 49.20 | 36.93 | 42.19 |
| CoCoA-Mix(ICML'25) | 73.09 | **74.97** | 74.01 | 91.04 | 77.37 | 83.64 | 90.09 | 90.93 | 90.50 | 33.51 | 34.15 | 33.83 |
| UPrompt(Ours) | 83.58 | 74.57 | **78.82** | 98.54 | **78.43** | **87.34** | **91.20** | **92.16** | **91.68** | **49.33** | **39.25** | **43.72** |

| Method | SUN397 | | | DTD | | | EuroSAT | | | UCF101 | | |
|---|---|---|---|---|---|---|---|---|---|---|---|---|
| | Base | Novel | HM | Base | Novel | HM | Base | Novel | HM | Base | Novel | HM |
| CoOp(IJCV'22) | 80.60 | 65.89 | 72.51 | 79.44 | 41.18 | 54.24 | 93.19 | 54.74 | 68.69 | 84.69 | 56.05 | 67.46 |
| PSRC(ICCV'23) | 82.67 | 78.47 | 80.52 | 83.37 | 62.97 | 71.75 | 92.90 | 73.90 | 82.32 | 87.10 | 78.80 | 82.74 |
| TAP(ICLR'25) | 82.87 | 79.53 | 81.17 | 84.20 | **68.00** | 75.24 | 90.70 | 82.17 | 86.22 | 87.90 | **82.43** | 85.08 |
| CLIP-AST(CVPR'25) | 83.05 | 78.12 | 80.51 | 84.03 | 65.34 | 73.52 | **95.90** | 81.72 | 88.24 | 87.38 | 79.12 | 83.05 |
| SurPL-G(ICML'25) | 83.43 | 78.96 | 81.13 | **86.07** | 62.04 | 72.11 | 94.63 | 81.33 | 87.48 | **89.44** | 79.74 | 84.31 |
| CoCoA-Mix(ICML'25) | 78.51 | 76.60 | 77.54 | 72.80 | 64.29 | 68.25 | 83.49 | 69.11 | 75.54 | 81.28 | 77.75 | 79.47 |
| UPrompt(Ours) | **83.77** | **80.05** | **81.87** | 85.60 | 67.23 | **75.31** | 94.82 | **82.68** | **88.33** | 89.21 | 81.83 | **85.36** |

and SurPL-G (81.03%) Liu et al., which generates diverse features across granularities, as well as latest methods like CLIP-AST (81.06%) Huang et al. (2025) and CoCoA-Mix (77.03%) Hong et al.. While TAP and SurPL-G also leverage multi-level representations, they treat different granularities as independent modules. In contrast, UPrompt's U-shaped architecture establishes a unified framework with bidirectional information flow across hierarchical levels, providing a more robust mechanism for generalizing from base to novel classes and achieving significant gains, including a +0.79% HM on the challenging FGVCAircraft dataset.

**Few-shot classification.** UPrompt achieves a leading 85.13% averaged accuracy across 11 datasets at 16-shot (Fig. 3), surpassing recent methods: GalLoP Lafon et al. (2024) (84.50%), which learns from separate global and local features; ProVP-Ref Xu et al. (2025) (83.07%), which builds progressive layer-wise connections; and MMA Yang et al. (2024) (82.76%), which adapts only higher-level representations. On ImageNet, our method performs best across all shots. Unlike these approaches lacking systematic cross-level interaction, UPrompt's bidirectional connection creates richer information flow across multi-granularity hierarchies, providing consistent gains from 1-shot to 16-shot.

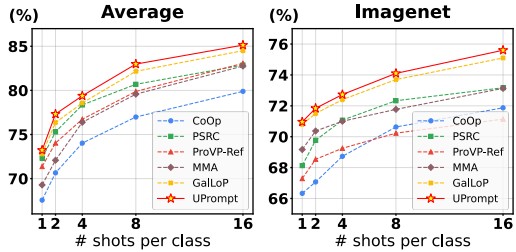

Figure 3: **Few-shot classification.** Average performance across 11 datasets and ImageNet results. Remaining results are in the Appendix B.1.

Table 3: **Out-of-distribution (OOD) generalization.** '*' means reproduced results. Best results highlighted in first, second.

| Method | Source | Target | | | | |
|---|---|---|---|---|---|---|
| | ImgNet | -V2 | -S | -A | -R | OOD |
| CoOp | 71.51 | 64.44 | 47.61 | 49.53 | 74.98 | 59.14 |
| PSRC | 71.27 | 64.35 | 49.55 | 50.90 | 77.80 | 60.65 |
| GalLoP* | 71.14 | 64.32 | 49.56 | 50.83 | 77.42 | 60.53 |
| SPTR | 70.05 | 64.40 | 48.78 | 51.30 | 77.90 | 60.59 |
| MMRL | 72.03 | 64.47 | 49.13 | 51.20 | 77.53 | 60.58 |
| HiCroPL | 71.22 | 64.33 | 49.47 | 50.79 | 77.15 | 60.44 |
| UPrompt | 72.25 | 65.06 | 50.43 | 51.33 | 78.04 | 61.22 |

**Out-of-distribution Generalization.** Domain shift evaluation examines semantic preservation (Table 3). GaLoP Lafon et al. (2024) sparse feature selection loses cross-domain information, SPTR Cui et al. (2025) optimal transport maintains single-granularity stability, MMRL Guo & Gu (2025) representation learning preserves generalization through decoupling, while HiCroPL Zheng et al. (2025) bidirectional refinement focuses on task-specific alignment rather than domain robustness (60.44%). UPrompt achieves 61.22% through multi-granularity architecture with cascaded enhancement providing global guidance and hierarchical supervision preventing semantic drift.

## 4.2 ABLATION STUDY

**Component Ablation.** We conduct an ablation study on Flickr30K (Table 4) to isolate the contributions of our core components: coarse-to-fine cascaded enhancement (CE) and fine-to-coarse hierarchical supervision (HS). We use the "Fine-grained only" model (92.2% I2T R@1) as our single-scale reference. Without CE, fine-grained features in multi-granularity architectures are very similar to single-granularity features. CE improves performance by maintaining global context while preserving fine details in "Vision only" mode (93.0% I2T R@1), providing progressive textual enhancement in "Text only" mode, and enabling comprehensive local relationship

Table 4: **Ablation study on Flickr30K.** CE: coarse-to-fine Cascaded Enhancement; HS: fine-to-coarse Hierarchical Supervision. Baseline uses original image-text pairs; Fine-grained only extends baseline with finest-granularity features; "gray" is our default set.

| Granularity | Method | Strategy | | I2T | | T2I | |
|---|---|---|---|---|---|---|---|
| | | CE | HS | R@1 | R@5 | R@1 | R@5 |
| Single | Baseline | ✗ | ✗ | 91.2 | 98.1 | 80.0 | 94.6 |
| | Fine-grained only | ✗ | ✗ | 92.2 | 98.3 | 81.1 | 95.2 |
| Multiple | (*Effectiveness of CE*) | | | | | | |
| | Vision only | ✓ | ✗ | 93.0 | 98.8 | 82.6 | 95.7 |
| | Text only | ✓ | ✗ | 92.7 | 98.6 | 82.4 | 95.5 |
| | Vision & Text | ✓ | ✗ | 93.3 | 99.1 | 83.0 | 95.9 |
| | (*Effectiveness of HS*) | | | | | | |
| | Coarse-to-fine | ✗ | ✓ | 86.8 | 95.1 | 75.3 | 91.5 |
| | Fine-to-coarse | ✗ | ✓ | 92.4 | 98.3 | 81.2 | 95.3 |
| | Full | ✓ | ✓ | **93.8** | **99.4** | **83.6** | **96.3** |

modeling under global guidance when combined for "Vision & Text" (93.3% I2T R@1). The directional nature of HS is critical; reversed coarse-to-fine supervision severely degrades performance by forcing detailed representations to model ambiguous signals. Conversely, fine-to-coarse hierarchical supervision in isolation offers negligible improvement, as enforcing semantic consistency at coarse levels lacks a mechanism to refine independently optimized fine-grained prompts. The full UPrompt model achieves best results (93.8% I2T R@1), revealing crucial connection where CE establishes cross-scale feature dependencies, allowing multi-level semantic consistency enforced by HS to effectively improve the entire representation hierarchy.

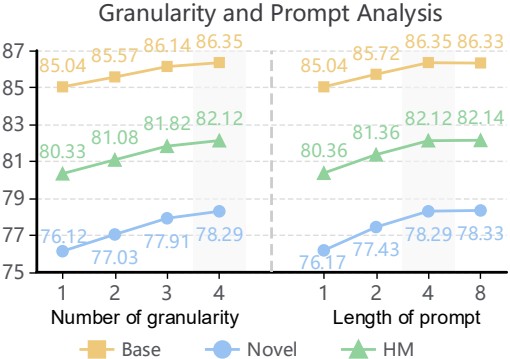

Figure 4: Granularity number and prompt length effects on classification. Left: varied granularity number (1-4). Right: varied prompt length (1-8).

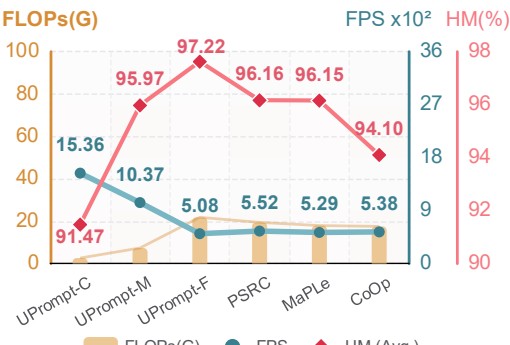

Figure 5: Efficiency-performance trade-offs across granularity levels. C, M, F denote coarse, medium, and fine configurations respectively.

**Effectiveness of Granularity Number and Prompt Length.** We analyze sensitivity of UPrompt to the number of granularity and prompt lengths (Fig. 4). Starting from the finest level and progressively adding coarser levels, performance consistently improves as the number increases from 1 to 4, with harmonic mean

Table 5: Granularity interval strategy comparison on Flickr30K I2T retrieval using R@1.

| Strategy | Coarse | Medium | Fine |
|---|---|---|---|
| $14 \times 14 \rightarrow 5 \times 5 \rightarrow 2 \times 2$ | 78.6 | 88.3 | 93.6 |
| $14 \times 14 \rightarrow 7 \times 7 \rightarrow 4 \times 4$ | 82.8 | 89.4 | 93.8 |

reaching 82.12% versus 80.33% at single finest granularity, though improvement rate gradually slows with increasing computational overhead. We therefore adopt 4 granularity levels balancing performance and efficiency. Beyond granularity number, we examine downsampling interval strategies. Following U-Net's $2\times2$ downsampling principle Ronneberger et al. (2015), we employ approximate halving of visual token resolution ($14 \times 14 \rightarrow 7 \times 7 \rightarrow 4 \times 4$ for retrieval). Table 5 compares this against a sparser interval ($14 \times 14 \rightarrow 5 \times 5 \rightarrow 2 \times 2$). Both achieve comparable fine-grained performance, but sparse intervals degrade at coarser levels due to reduced expressive capacity, demonstrating the effectiveness of our design. For prompt length, performance in Fig. 4 (right) peaks at length 4 and remains stable at length 8, thus we set prompt length to 4.

**Analysis of Performance-Cost.** Fig. 5 reveals the performance and cost across different granularities, where UPrompt-C, UPrompt-M, and UPrompt-F are coarse ($1 \times 1$), medium ($7 \times 7$), and fine ($14\times14$) visual tokens with corresponding textual granularities. UPrompt-F substantially outperforms existing prompt learning methods with 97.22% average HM across OxfordPets and Caltech101 with limited additional cost. UPrompt-M achieves comparable performance (95.97% average HM) and matches PSRC's accuracy using only 1/3 of PSRC's computational cost. UPrompt-C requires minimal resources while preserving reasonable performance at 91.47% average HM. This flexibility stems from our architecture where fine-to-coarse hierarchical supervision enables coarse levels to benefit from detailed representations, allowing adaptive granularity selection based on resource constraints.

**Robustness to Different LLMs.** We evaluate the sensitivity of our text hierarchy generation to different LLMs by conducting experiments with Qwen3-4B, Qwen3-14B Yang et al. (2025) and Llama3-8B on Flickr30K cross-modal retrieval (Table 6). The performance remains stable across different models with varying architectures and scales (4B to 14B parameters), confirming that our method is not sensitive to the specific LLM used for text hierarchy generation.

Table 6: Text hierarchy robustness across different LLMs on Flickr30K.

| Models | Image-to-text | | | Text-to-image | | |
|---|---|---|---|---|---|---|
| | R@1 | R@5 | R@10 | R@1 | R@5 | R@10 |
| Qwen3-4B | 93.4 | 99.4 | 99.5 | 83.4 | 96.2 | 98.6 |
| Llama 3-8B | 93.8 | 99.4 | 99.6 | 83.6 | 96.3 | 98.4 |
| Qwen3-14B | 93.7 | 99.5 | 99.8 | 83.9 | 96.2 | 98.7 |

**Analysis of Bidirectional Information Flow.** We conduct an ablation study on Flickr30K to isolate our bidirectional connection components, with results in Table 7. Fine-to-coarse Hierarchical Supervision (HS) substantially boosts coarse-grained performance (I2T R@1 from 75.4% to 82.8%), addressing the semantic drift caused by optimizing on the ambiguous signals inherent in simplified representations. Coarse-to-fine Cascaded Enhancement (CE) improves fine-grained performance (I2T R@1 from 92.7% to 93.8%), resolving context deficiency from isolated local detail modeling, without an understanding of their role within the global scene. Both components are integral: HS maintains semantic consistency for coarse representations, while CE provides contextual guidance for fine ones. To verify fine-grained supervision reliability, we compared single fine-layer supervision against mixed fine and medium-layer supervision (Appendix B.7). Results confirm fine-layer supervision alone achieves comparable performance, validating its sufficiency as the teacher signal.

Table 7: **Effectiveness of Cascaded Enhancement (CE) and Hierarchical Supervision (HS)** on cross-modal retrieval on Flickr30K with different granularity sets.

| Granularity | Method | Image-to-text | | | Text-to-image | | |
|---|---|---|---|---|---|---|---|
| | | R@1 | R@5 | R@10 | R@1 | R@5 | R@10 |
| Coarse-grained | w/o HS | 75.4 | 89.9 | 92.8 | 60.4 | 82.3 | 86.8 |
| | w/ HS | 82.8 | 94.4 | 93.5 | 67.6 | 87.5 | 90.3 |
| Fine-grained | w/o CE | 92.7 | 98.5 | 99.2 | 81.7 | 95.8 | 98.0 |
| | w/ CE | 93.8 | 99.4 | 99.6 | 83.6 | 96.3 | 98.4 |

**Resolution Analysis.** To validate that multi-granularity improvements stem from semantic hierarchy rather than token density compensation, we conduct resolution-scale ablation on Flickr30K (Table 8). Results show that increasing the granularity level (Level 1→3) yields substantial gains at both 224×224 (82.8%→93.8%) and 336×336

Table 8: **Resolution×granularity level ablation** on Flickr30K (I2T R@1). Starting from the coarsest level, we progressively add finer granularity levels.

| Resolution | Level 1 | Level 2 | Level 3 |
|---|---|---|---|
| 224×224 | 82.8 | 89.4 | 93.8 |
| 336×336 | 83.3 | 92.2 | 95.1 |

(83.3%→95.1%), while resolution alone improves marginally (82.8%→83.3%). This demonstrates multi-granularity benefits dominate across resolutions, confirming our method addresses semantic hierarchy independent of spatial token density. Appendix B.4 further validates consistent improvements with denser backbones (ViT-L), demonstrating framework generalization across model scales.

| Original image-text pair | Visual Tokens: 4×4 | Visual Tokens: 7×7 | Visual Tokens: 14×14 |
|---|---|---|---|
| 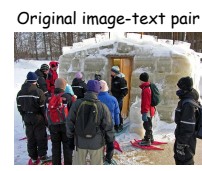 | 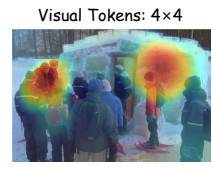 | 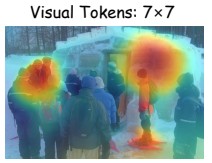 | 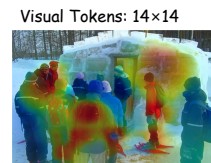 |
| *A group of people wearing snowshoes, and dressed for winter hiking, is standing in front of a building that looks like it's made of blocks of ice.* | *Group in winter gear stands in front of icy building blocks.* | *A group of people wearing snowshoes, and dressed for winter hiking, is standing in front of a building that looks like it's made of blocks of ice.* | *A group of people, dressed in winter gear like beanies, skiing jackets, gloves, and backpacks, stand quietly outside an igloo made of blocks of ice, listening intently as someone explains the story of this frozen abode.* |

Figure 6: **Multi-granularity visual attention visualization.** Heat maps show bidirectional connection effects with cascaded enhancement preserving global context in fine-grained attention while hierarchical supervision reduces semantic drift across different levels.

### 4.3 VISUALIZATION

**Visualization of Multi-Granularity Attention.** Fig. 6 validates UPrompt's bidirectional connection. Cascaded Enhancement enables fine-grained attention (14×14) to maintain global coherence while capturing details like winter gear and textures. Hierarchical Supervision ensures coarser levels (4×4, 7×7) focus on semantically relevant regions, preventing background noise interference and semantic drift. The progressive refinement from global to local demonstrates effective multi-scale integration, where each granularity captures complementary information while maintaining semantic consistency. Fig. 8 in Appendix B.8 further shows our multi-granularity design across retrieval.

**Visualization of Bidirectional Connection Components.** Fig. 9 and Fig. 10 in Appendix B.9, B.10 visualize Coarse-to-Fine Enhancement injecting global context into fine-grained representations and Fine-to-Coarse Supervision leveraging finest-level alignment to regularize coarser granularities.

## 5 CONCLUSION

In this work, we present UPrompt, a simple yet effective framework that addresses the limitation of single granularity in vision-language prompt learning. Inspired by U-Net, our method constructs parallel multi-granularity representations with bidirectional connections to facilitate information flow across scales. This consists of coarse-to-fine enhancement that injects global context into local details, and fine-to-coarse supervision that ensures semantic consistency. Extensive experiments demonstrate effectiveness across cross-modal retrieval, base-to-novel generalization, and few-shot classification. Despite its effectiveness, UPrompt has limitations in that the granularity hierarchy requires manual design. Future work will explore adaptive granularity selection based on task complexity.

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

# Appendix

## A PROOF FOR THE PROPOSITION

### A.1 PROOF FOR PROPOSITION 1

**Proposition 1** (CE Directional Alignment Effect). *Let $\hat{X}^{(k)}$ be the fine-grained representation at level k enhanced by coarse-to-fine cascaded enhancement (CE, Eq. (5)–(6)), which leverages contextual guidance from the coarser representation $\hat{X}^{(k-1)}$. Let $X^{(k)}$ be its unenhanced counterpart. Under the mild assumption that the coarse context is informative, CE provably strengthens the alignment between fine-grained features and their coarse-grained guidance in expectation:*

$$\mathbb{E}\left[\frac{\langle \hat{X}^{(k)}, \hat{X}^{(k-1)} \rangle}{\|\hat{X}^{(k)}\| \, \|\hat{X}^{(k-1)}\|}\right] \;\geq\; \mathbb{E}\left[\frac{\langle X^{(k)}, \hat{X}^{(k-1)} \rangle}{\|X^{(k)}\| \, \|\hat{X}^{(k-1)}\|}\right].$$

**Proof.** Let $u \triangleq \hat{X}^{(k-1)}/\|\hat{X}^{(k-1)}\|$ be the unit coarse direction. Denote element-wise absolute value by $|\cdot|$, and define the "sign-adjusted" vector $v \in \mathbb{R}^d$ by $v_i \triangleq \text{sign}(X_i^{(k)} u_i)\,|u_i|$. Then for any nonnegative gate $a \in \mathbb{R}_{\geq 0}^d$ we can rewrite the cosine with $u$ as

$$\frac{\langle X^{(k)} \odot a, \, u \rangle}{\|X^{(k)} \odot a\|} \;=\; \frac{\langle |X^{(k)}| \odot a, \, v \rangle}{\| \, |X^{(k)}| \odot a \, \|}.$$

Under CE (Eq. (5)–(6)), the enhanced representation takes the form $\hat{X}^{(k)} = X^{(k)} \odot A\big(X^{(k)}, \hat{X}^{(k-1)}\big)$ with an *element-wise* nonnegative map $A$ produced from cross-granularity softmax attention and a value mixing of $\hat{X}^{(k-1)}$; we treat $a = A\big(X^{(k)}, \hat{X}^{(k-1)}\big)$ as the random gate induced by CE.

**Monotone informative-gate (MIG) assumption** Park et al. (2019). Formally, we instantiate the "coarse context is informative" condition as:

$$\mathbb{E}\big[a_i \,\big|\, X^{(k)}, u\big] \text{ is coordinatewise nondecreasing in } r_i \triangleq \frac{v_i}{|X_i^{(k)}|}, \tag{12}$$

i.e., coordinates that are better aligned with $u$ (larger $r_i$) receive, in expectation, larger CE weights.[1]

**Directional-derivative lemma.** Define $f(a) \triangleq \dfrac{\langle |X^{(k)}| \odot a, \, v \rangle}{\| \, |X^{(k)}| \odot a \, \|}$. A direct calculation gives, for any coordinate $i$,

$$\frac{\partial f}{\partial a_i}(a) \;=\; \frac{|X_i^{(k)}| \, v_i}{\| \, |X^{(k)}| \odot a \, \|} \;-\; f(a) \cdot \frac{a_i \, |X_i^{(k)}|^2}{\| \, |X^{(k)}| \odot a \, \|}.$$

Evaluated at the identity gate $a = \mathbf{1}$, letting $f_0 \triangleq f(\mathbf{1}) = \dfrac{\langle |X^{(k)}|, v \rangle}{\|X^{(k)}\|}$, we obtain the directional derivative along any perturbation $w$:

$$\frac{d}{d\alpha} f(\mathbf{1} + \alpha w)\bigg|_{\alpha=0} \;=\; \frac{1}{\|X^{(k)}\|} \sum_{i=1}^{d} w_i \, |X_i^{(k)}| \left(\frac{v_i}{|X_i^{(k)}|} - f_0\right) \;=\; \frac{1}{\|X^{(k)}\|} \sum_{i=1}^{d} w_i \, |X_i^{(k)}| \, (r_i - f_0).$$

Hence the first-order increase of $f$ at $\mathbf{1}$ is nonnegative whenever $w$ is (on average) positively associated with the alignment score $r$.

**Homotopy argument.** Consider the linear path $a(t) = \mathbf{1} + t\,(a - \mathbf{1})$ for $t \in [0, 1]$. Differentiating along the path,

$$\frac{d}{dt} f\big(a(t)\big) \;=\; \sum_{i=1}^{d} (a_i - 1) \, \frac{\partial f}{\partial a_i}\big(a(t)\big).$$

---

[1]This matches the CE mechanism: the softmax attention in Eq. (6) increases weights where the local embedding is more similar to the coarse context, and the resulting map gates $X^{(k)}$ element-wise in Eq. (5); see also the need to make such conditions explicit when CE is *element-wise* rather than additive.

By continuity of $\frac{\partial f}{\partial a_i}$ and the preceding lemma, it suffices that the *expected* increment $(a_i - 1)$ remain positively associated with the local alignment score along the path. The MIG assumption (12) guarantees exactly this: conditioned on $(X^{(k)}, u)$, the coordinates with larger $r_i$ receive larger expected weights at every $t$, so $\mathbb{E}\left[\frac{d}{dt} f\big(a(t)\big) \mid X^{(k)}, u\right] \geq 0$ for all $t \in [0, 1]$. Integrating from $t = 0$ to $t = 1$ yields

$$\mathbb{E}\big[f(a) \mid X^{(k)}, u\big] \;\geq\; f(\mathbf{1}) \;=\; \frac{\langle X^{(k)}, u \rangle}{\|X^{(k)}\|}.$$

Finally, taking expectation over $(X^{(k)}, \hat{X}^{(k-1)})$ proves

$$\mathbb{E}\left[ \frac{\langle \hat{X}^{(k)}, \hat{X}^{(k-1)} \rangle}{\|\hat{X}^{(k)}\| \, \|\hat{X}^{(k-1)}\|} \right] \;\geq\; \mathbb{E}\left[ \frac{\langle X^{(k)}, \hat{X}^{(k-1)} \rangle}{\|X^{(k)}\| \, \|\hat{X}^{(k-1)}\|} \right].$$

$\square$

**Remarks.** (i) The proof explicitly uses CE's *element-wise* form (5)–(6); additive/value-replacement assumptions are unnecessary. (ii) The MIG condition is a precise, verifiable sufficient condition tailored to element-wise gating, addressing the need to clarify when an element-wise CE improves directional alignment (and avoiding overly strong orthogonal-leakage assumptions).

### A.2  PROOF OF PROPOSITION 2

**Proposition 2** (HS Consistency and Substitutability)**.** *Let $S^{(k)}$ and $S^{(K)}$ be similarity matrices from Eq. (8), and define $p_{\tau_d}^{(k)}(j|i) = softmax(S_{i,:}^{(k)}/\tau_d)_j$ and $q_{\tau_d}^{(K)}(j|i) = softmax(S_{i,:}^{(K)}/\tau_d)_j$ where teacher $q^{(K)}$ is detached as in Eq. (9). Assuming HS aligns coarse-grained distributions with fine-grained teachers, HS bounds semantic drift and enables performance-preserving coarse inference:*

$$\mathbb{E}_{(x,t),i}\left[ \mathrm{KL}\Big( q_{\tau_d}^{(K)}(\cdot|i) \,\|\, p_{\tau_d}^{(k)}(\cdot|i) \Big) \right] \leq \varepsilon \implies \mathbb{E}_{(x,t),i}\big[\big| \Phi\big(p_{\tau_d}^{(k)}(\cdot|i)\big) - \Phi\big(q_{\tau_d}^{(K)}(\cdot|i)\big) \big|\big] \leq L\sqrt{\varepsilon/2}$$

(13)

*for any $L$-Lipschitz functional $\Phi$ w.r.t. total variation distance. The detach operation ensures gradient isolation: $\partial L_{guide}/\partial z^{(K)} = 0$.*

**Proof.**  Fix $(x, t)$ and anchor index $i$, and set $Q := q_{\tau_d}^{(K)}(\cdot \mid i)$ and $P := p_{\tau_d}^{(k)}(\cdot \mid i)$. By Pinsker's inequality,

$$\mathrm{TV}(P, Q) \;\leq\; \sqrt{\tfrac{1}{2}\,\mathrm{KL}(Q \,\|\, P)}.$$

For any functional $\Phi$ that is $L$-Lipschitz w.r.t. total variation,

$$\big|\Phi(P) - \Phi(Q)\big| \;\leq\; L\,\mathrm{TV}(P, Q) \;\leq\; L\sqrt{\tfrac{1}{2}\,\mathrm{KL}(Q \,\|\, P)}.$$

Taking expectation over $(x, t), i$ and applying Jensen's inequality (since $\sqrt{\cdot}$ is concave) yields

$$\mathbb{E}_{(x,t),i}\big[\big|\Phi(P) - \Phi(Q)\big|\big] \;\leq\; L\,\mathbb{E}\left[\sqrt{\tfrac{1}{2}\,\mathrm{KL}(Q \,\|\, P)}\right] \;\leq\; L\sqrt{\tfrac{1}{2}\,\mathbb{E}[\mathrm{KL}(Q \,\|\, P)]} \;\leq\; L\sqrt{\varepsilon/2},$$

which proves the stated consistency/substitutability bound.

For the gradient isolation, write the HS guidance loss as

$$\mathcal{L}_{\text{guide}} \;=\; \mathbb{E}_{(x,t),i}\left[\mathrm{KL}\Big(q_{\tau_d}^{(K)}(\cdot \mid i) \,\|\, p_{\tau_d}^{(k)}(\cdot \mid i)\Big)\right],$$

where the teacher $q_{\tau_d}^{(K)}$ is detached as in Eq. (8). Hence $q_{\tau_d}^{(K)}$ is treated as a constant and

$$\frac{\partial \mathcal{L}_{\text{guide}}}{\partial z^{(K)}} = 0.$$

Equivalently, gradients flow only to the coarse head via the similarity logits $S^{(k)}$ from Eq. (7): if $P = \mathrm{softmax}(S_{i,:}^{(k)}/\tau_d)$, then

$$\frac{\partial \mathcal{L}_{\text{guide}}}{\partial S_{i,:}^{(k)}} \;=\; \frac{1}{\tau_d}\big(P - Q\big),$$

which is the standard soft-target distillation gradient scaled by $1/\tau_d$. $\square$

# B  OTHER EXPERIMENTAL RESULTS

## B.1  INDIVIDUAL DATASET FEW-SHOT CLASSIFICATION

Results in Figure 7 confirm UPrompt's effectiveness across diverse domains. On fine-grained tasks like StanfordCars and FGVCAircraft, our method outperforms CoOp and PSRC by capturing both specific details and global patterns through multi-granularity representations. For DTD texture classification, UPrompt surpasses MMA and ProVP-Ref via bidirectional connection that enables fine-grained modeling guided by coarse context. On SUN397 and EuroSAT, hierarchical supervision prevents semantic drift while maintaining competitive performance with GalLoP. Consistent improvements across shot configurations validate that our U-shaped architecture addresses granularity trade-offs in single-scale methods.

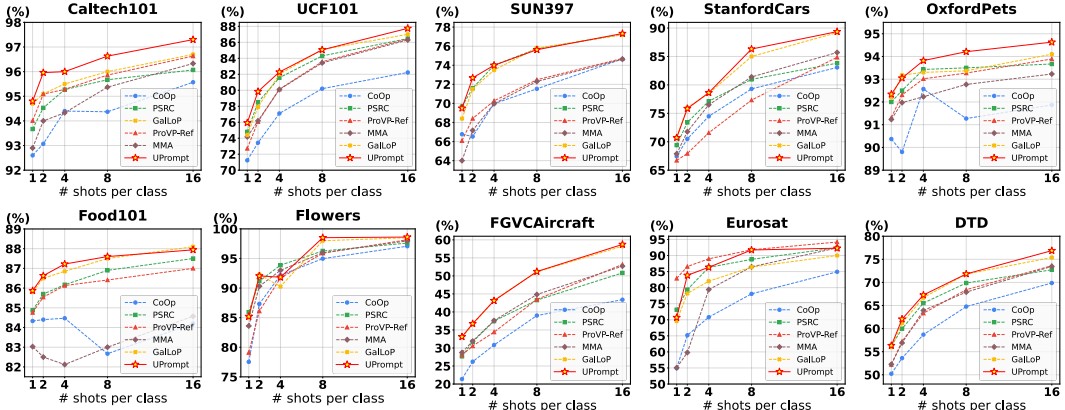

Figure 7: **Few-shot classification results on individual datasets.** Detailed performance breakdown across 10 evaluation datasets with 1, 2, 4, 8, and 16 shots per class.

Table 9: **Cross-dataset evaluation.** Domain transfer against recent prompt learning methods. Trained on ImageNet, evaluated on 10 datasets. Best results highlighted in first , second .

| | Source | Target | | | | | | | | | | |
|---|---|---|---|---|---|---|---|---|---|---|---|---|
| | ImgNet | Cal101 | Pets | Cars | Flowers | Food | FGVC | SUN | DTD | SAT | UCF | Avg. |
| CoOp(IJCV'22) | 71.51 | 93.70 | 89.14 | 64.51 | 68.71 | 85.30 | 18.47 | 64.15 | 41.92 | 46.39 | 66.55 | 63.88 |
| PSRC(ICCV'23) | 71.27 | 93.60 | 90.25 | 65.70 | 70.25 | 86.15 | 23.90 | 67.10 | 46.87 | 45.50 | 68.75 | 65.81 |
| DeKgTCP(ICLR'25) | 72.33 | 94.73 | 90.02 | 65.49 | 72.39 | 86.59 | 25.05 | 67.19 | 44.47 | 51.37 | 68.78 | 66.61 |
| TAP(ICLR'25) | 72.30 | 94.30 | 90.70 | 65.60 | 70.93 | 86.10 | 24.57 | 68.30 | 50.20 | 46.00 | 68.90 | 66.56 |
| TAC(CVPR'25) | 72.77 | 94.53 | 90.67 | 65.30 | 72.20 | 85.83 | 23.53 | 67.63 | 47.57 | 48.07 | 70.00 | 66.53 |
| HiCroPL(ICCV'25) | 70.84 | 94.48 | 90.13 | 65.68 | 72.03 | 86.46 | 26.58 | 68.78 | 53.19 | 49.19 | 70.31 | 67.38 |
| CoCoA-Mix(ICML'25) | 70.85 | 93.46 | 89.07 | 65.59 | 68.72 | 85.78 | 24.10 | 63.61 | 46.41 | 48.18 | 67.78 | 65.27 |
| UPrompt(Ours) | 72.25 | 94.75 | 90.97 | 66.09 | 72.41 | 86.60 | 26.44 | 68.64 | 47.51 | 52.08 | 70.05 | 67.55 |

## B.2  CROSS-DATASET EVALUATION

As presented in Table 9, our UPrompt framework demonstrates robust domain generalization capabilities when transferred from ImageNet to 10 downstream datasets. It achieves the highest average accuracy of 67.55%, underscoring the effectiveness of its architecture in adapting to new data distributions. We compare UPrompt with other methods that also leverage multi-level or hierarchical representations. For instance, TAP Ding et al. (2024) constructs an explicit "concept-attribute-description" hierarchy, while HiCroPL Zheng et al. (2025) establishes knowledge flow across network layers. Although these approaches are competitive, particularly HiCroPL on datasets like FGVC and SUN, our UPrompt's U-Net-inspired bidirectional multi-granularity learning leads to more consistent and superior performance across a wider range of tasks, securing the top results on 6 of the 10

Table 10: **Error bar analysis on cross-dataset evaluation.** Trained on ImageNet, evaluated on 10 datasets. Results report mean accuracy and standard deviation across three independent runs.

| Method | Caltech101 | OxfordPets | StanfordCars | Flowers102 | Food101 |
|---|---|---|---|---|---|
| CoOp | 93.70 | 89.14 | 64.51 | 68.71 | 85.30 |
| UPrompt | 94.51±0.11 | 90.75±0.17 | 65.98±0.08 | 72.19±0.23 | 86.57±0.03 |

| Method | FGVCAircraft | SUN397 | DTD | EuroSAT | UCF101 |
|---|---|---|---|---|---|
| CoOp | 18.47 | 64.15 | 41.92 | 46.39 | 66.55 |
| UPrompt | 26.14±0.13 | 68.42±0.21 | 47.35±0.08 | 53.24±1.18 | 69.93±0.20 |

target datasets. Furthermore, UPrompt outperforms other recent domain generalization methods like TAC Hao et al. (2025) and DeKgTCP Li et al. (2025), validating that our explicit modeling of coarse-to-fine semantic levels is highly effective for robust cross-dataset transfer.

## B.3 ERROR BAR ANALYSIS

We conducted an error bar analysis across 10 target datasets in the cross-dataset evaluation setting (trained on ImageNet), performing three independent runs to ensure robust statistical evaluation. The results in Table 10 report mean accuracy and standard deviation for both CoOp and UPrompt. UPrompt demonstrates remarkable stability across most datasets, with particularly low variance on Food101 (±0.03), StanfordCars (±0.08), and DTD (±0.08), indicating highly consistent cross-domain generalization. Even on challenging datasets like FGVCAircraft (±0.13) and UCF101 (±0.20), the variance remains low, validating the robustness of our bidirectional multi-granularity framework. The error bar analysis confirms that UPrompt provides reliable and consistent performance across diverse domain transfer scenarios.

Table 11: **Cross-modal retrieval results on ViT-B/32 backbone.** rSum is the sum of all R@1, R@5, and R@10 scores. Best results highlighted in first, second.

| Methods | Flickr30K | | | | | | | MSCOCO | | | | | | |
|---|---|---|---|---|---|---|---|---|---|---|---|---|---|---|
| | Image-to-Text | | | Text-to-Image | | | rSum | Image-to-Text | | | Text-to-Image | | | rSum |
| | R@1 | R@5 | R@10 | R@1 | R@5 | R@10 | | R@1 | R@5 | R@10 | R@1 | R@5 | R@10 | |
| MAMET(TCSVT'25) | 87.7 | 97.5 | 99.6 | 73.5 | 93.0 | 96.5 | 547.8 | 61.5 | 86.2 | 92.5 | 48.6 | 76.3 | 85.3 | 450.4 |
| APSE-IPIK(AAAI'25) | 86.3 | 97.6 | 99.4 | 72.0 | 92.5 | 95.1 | 542.9 | 59.1 | 85.7 | 94.6 | 45.1 | 72.8 | 82.5 | 439.8 |
| UPrompt(Ours) | 88.9 | 97.6 | 99.7 | 74.0 | 93.2 | 96.8 | 550.2 | 62.4 | 86.8 | 93.8 | 49.7 | 76.9 | 85.7 | 455.3 |

## B.4 CROSS-MODAL RETRIEVAL WITH DIFFERENT BACKBONES

Table 12: **Image-to-text retrieval on ViT-L/14 backbone.** Results highlighted in first, second.

| Methods | MSCOCO | | | Flickr30K | | | rSum |
|---|---|---|---|---|---|---|---|
| | R@1 | R@5 | R@10 | R@1 | R@5 | R@10 | |
| Unicoder-VL | 62.3 | 87.1 | 92.8 | 86.2 | 96.3 | 99.0 | 523.7 |
| Oscar | 73.5 | 92.2 | 96.0 | - | - | - | - |
| ERNIE-ViL | - | - | - | 88.7 | 98.0 | 99.2 | - |
| AAPE | 76.7 | 94.5 | 97.4 | 94.9 | 99.3 | 99.7 | 561.8 |
| UPrompt | 77.8 | 94.9 | 97.4 | 95.1 | 99.7 | 99.8 | 564.7 |

To validate UPrompt's generalizability across different architectures, we evaluate on ViT-B/32 and ViT-L/14 backbones (Tables 11 and 12). On ViT-B/32, UPrompt achieves 550.2 and 455.3 rSum on Flickr30K and MSCOCO respectively, outperforming MAMET Wang et al. (2025b) (547.8, 450.4) and APSE-IP1K Huang et al. (2025) (542.9, 439.8). The consistent improvements across different model scales demonstrate the robustness of our multi-granularity framework. On the larger ViT-L/14 backbone for image-to-text retrieval, UPrompt achieves 564.7 rSum, outperforming AAPE Huang et al. (2024) (561.8), Unicoder-VL Li et al. (2020a) (523.7), Oscar Li et al. (2020b), and ERNIE-ViL Yu et al. (2021), with particularly strong performance on Flickr30K (95.1% R@1) and MSCOCO (77.8% R@1). These results confirm that our bidirectional connection mechanisms effectively leverage increased model capacity, with hierarchical supervision preventing semantic drift across granularities regardless of backbone architecture.

Table 13: **Base-to-novel generalization on alternative VLM architectures.** UPrompt consistently outperforms baselines across SigLIP and EVA-CLIP.

| Methods | Backbone | Cars | Flowers | FGVC |
|---------|----------|------|---------|------|
| CoOp | EVA-CLIP | 71.33 | 77.39 | 34.72 |
| UPrompt | EVA-CLIP | **79.42** | **87.36** | **43.86** |
| CoOp | SigLIP | 92.33 | 89.42 | 38.27 |
| UPrompt | SigLIP | **94.67** | **93.26** | **46.34** |

Table 14: **Rule-based hierarchies** on CUB-200 and AWA2 in generalized zero-shot learning.

| Dataset | Method | Level | Base | New | HM |
|---------|--------|-------|------|-----|-----|
| AWA2 | CoOp | Single | 95.32 | 72.68 | 82.47 |
| | Uprompt | Level 1 | 93.24 | 70.27 | 80.14 |
| | | Level 2 | 95.81 | 73.13 | 82.95 |
| | | Level 3 | **96.70** | **74.62** | **84.24** |
| CUB-200 | CoOp | Single | 63.78 | 49.23 | 55.57 |
| | UPrompt | Level 1 | 61.36 | 46.84 | 53.13 |
| | | Level 2 | 64.65 | 50.42 | 56.65 |
| | | Level 3 | **66.12** | **51.26** | **57.75** |

Table 15: **Fine-Layer Supervision Reliability.** Comparison of **fine-layer** versus **fine + medium-layer** supervision on Flickr30K and MSCOCO. rSum denotes sum of all R@1, R@5, R@10 scores.

| Teacher Strategy | Flickr30K | | | | | | | MSCOCO | | | | | | |
|------------------|-----------|---|---|---|---|---|------|--------|---|---|---|---|---|------|
| | Image-to-Text | | | Text-to-Image | | | | Image-to-Text | | | Text-to-Image | | | |
| | R@1 | R@5 | R@10 | R@1 | R@5 | R@10 | rSum | R@1 | R@5 | R@10 | R@1 | R@5 | R@10 | rSum |
| Fine + Medium (soft mixing) | 93.7 | 99.2 | 99.3 | 83.9 | 96.3 | 98.2 | 570.6 | 69.8 | 89.4 | 94.5 | 52.5 | 78.8 | 87.5 | 472.5 |
| Fine only (Ours) | 93.8 | 99.4 | 99.6 | 83.6 | 96.3 | 98.4 | 571.1 | 70.1 | 89.8 | 84.8 | 52.6 | 79.1 | 87.9 | 474.3 |

## B.5 GENERALIZATION ACROSS VLM ARCHITECTURE

To validate UPrompt's generalizability across diverse vision-language models, we evaluate on alternative architectures including SigLIP and EVA-CLIP with ViT-B/16 backbones. Table 13 presents base-to-novel generalization results on three representative datasets. UPrompt achieves consistent improvements over CoOp across both architectures: on EVA-CLIP, gains of +8.09%, +9.97%, and +9.14% HM on StanfordCars, Flowers102, and FGVCAircraft respectively; on SigLIP, gains of +2.34%, +3.84%, and +8.07% HM respectively. Notably, the improvements are particularly pronounced on the fine-grained FGVCAircraft dataset (+9.14% on EVA-CLIP, +8.07% on SigLIP), demonstrating that our multi-granularity framework effectively enhances fine-grained recognition across different VLM architectures. These results confirm that our bidirectional connection mechanisms operate effectively at the prompt level, independent of the underlying vision-language model architecture.

## B.6 RULE-BASED TEXT CONSTRUCTION.

To demonstrate that our framework's effectiveness does not rely on LLM-generated priors, we conduct experiments using rule-based text construction on CUB-200 Wah et al. (2011) and AWA2 Xian et al. (2017) datasets, which provide dense attribute annotations. We construct three granularity levels: Level 1 (coarse) uses "a photo of a class", Level 2 (medium) adds the highest-certainty attribute (e.g., "a photo of a class with black bill"), and Level 3 (fine) incorporates two highest-certainty attributes (e.g., "a photo of a class with black bill and white breast"), where attributes are ranked by certainty scores from dataset metadata. Visually, the three levels correspond to $4{\times}4$ pooled tokens, $7{\times}7$ pooled tokens, and the original $14{\times}14$ tokens respectively. We compare against CoOp, which uses "a photo of a class" with $14{\times}14$ visual tokens as the single-granularity baseline. Table 14 presents results on generalized zero-shot learning (GZSL). The progressive improvements from coarse to fine levels validate that our bidirectional connection mechanism effectively integrates multi-scale representations, even without external knowledge sources like LLM generated content.

## B.7 RELIABILITY OF FINE-GRAINED SUPERVISION.

To validate that fine-grained representations provide reliable supervision signals for coarser levels, we compared single fine-layer supervision with mixed supervision combining fine and medium-granularity teachers on Flickr30K and MSCOCO. Results in Table 15 demonstrate that fine-layer supervision alone achieves comparable or superior performance across both datasets (Flickr30K I2T R@1: 93.8% vs 93.7%; MSCOCO I2T R@1: 70.1% vs 69.8%), confirming its sufficiency and stability as the primary teacher signal without requiring multi-layer aggregation.

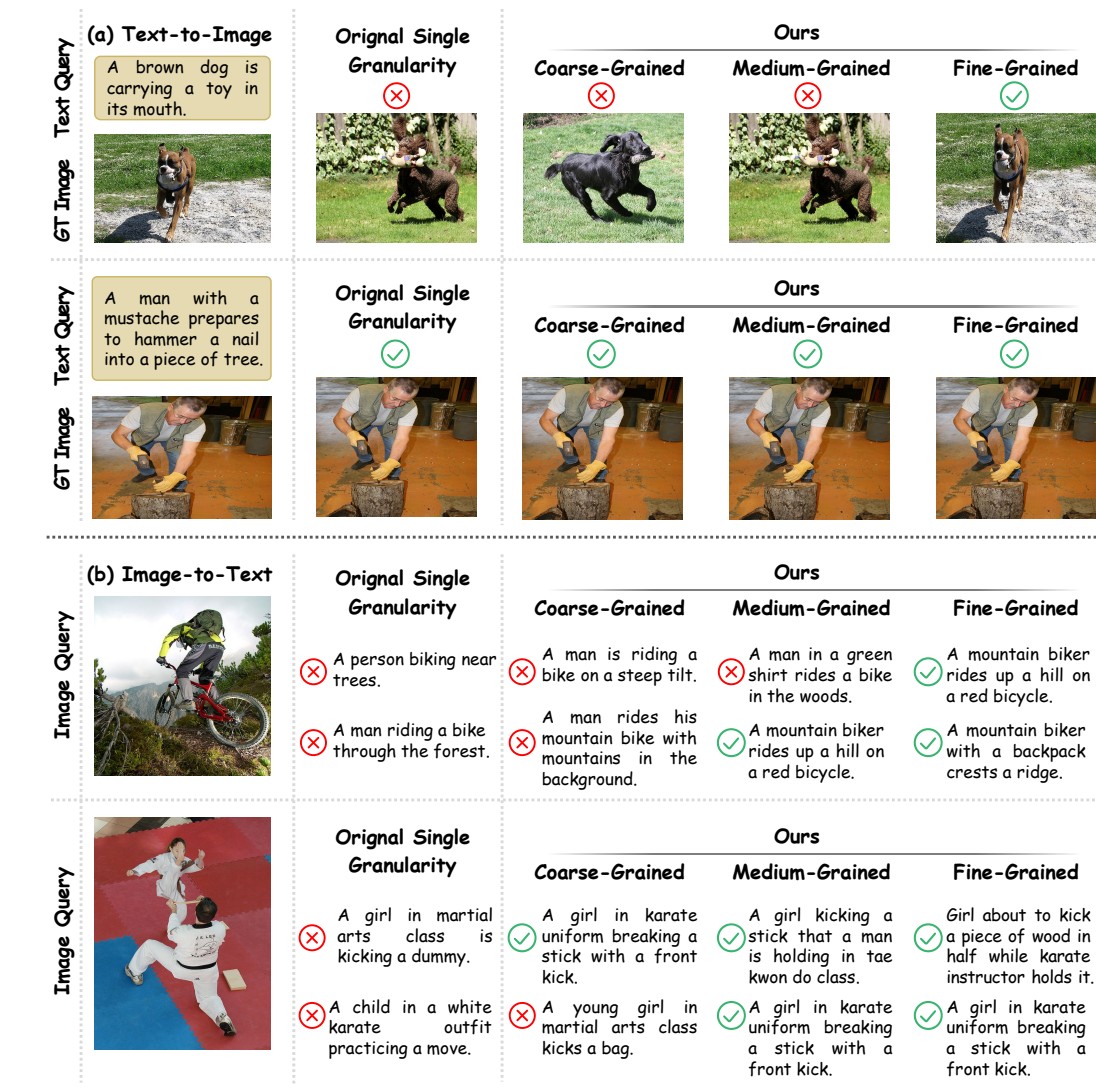

Figure 8: **Cross-modal retrieval results on Flickr30K dataset.** "Original Single Granularity" refers to baseline model using fixed single-scale visual and textual representations. ⊗ indicate retrieval failures, ✓ indicate successful retrievals.

## B.8    GRANULARITY-SPECIFIC RETRIEVAL EFFECTIVENESS.

The retrieval results in Fig. 8 validate our bidirectional connection mechanisms across granularity levels. Fine-grained representations consistently excel in both Text-to-Image and Image-to-Text tasks, resolving challenging cases requiring precise semantic understanding, such as distinguishing "carrying a toy in its mouth" from general dog activities. This capability stems from cascaded enhancement providing global contextual guidance, preventing attention from focusing solely on isolated details while capturing comprehensive information. Medium-grained representations outperform the single-granularity baseline while using fewer visual tokens. Coarse-grained representations achieve comparable performance despite using substantially fewer visual and textual tokens, enabled by hierarchical supervision that prevents semantic drift and preserves alignment quality with reduced representational capacity. These results confirm flexible performance-efficiency trade-offs while keeping semantic consistency across hierarchical structure.

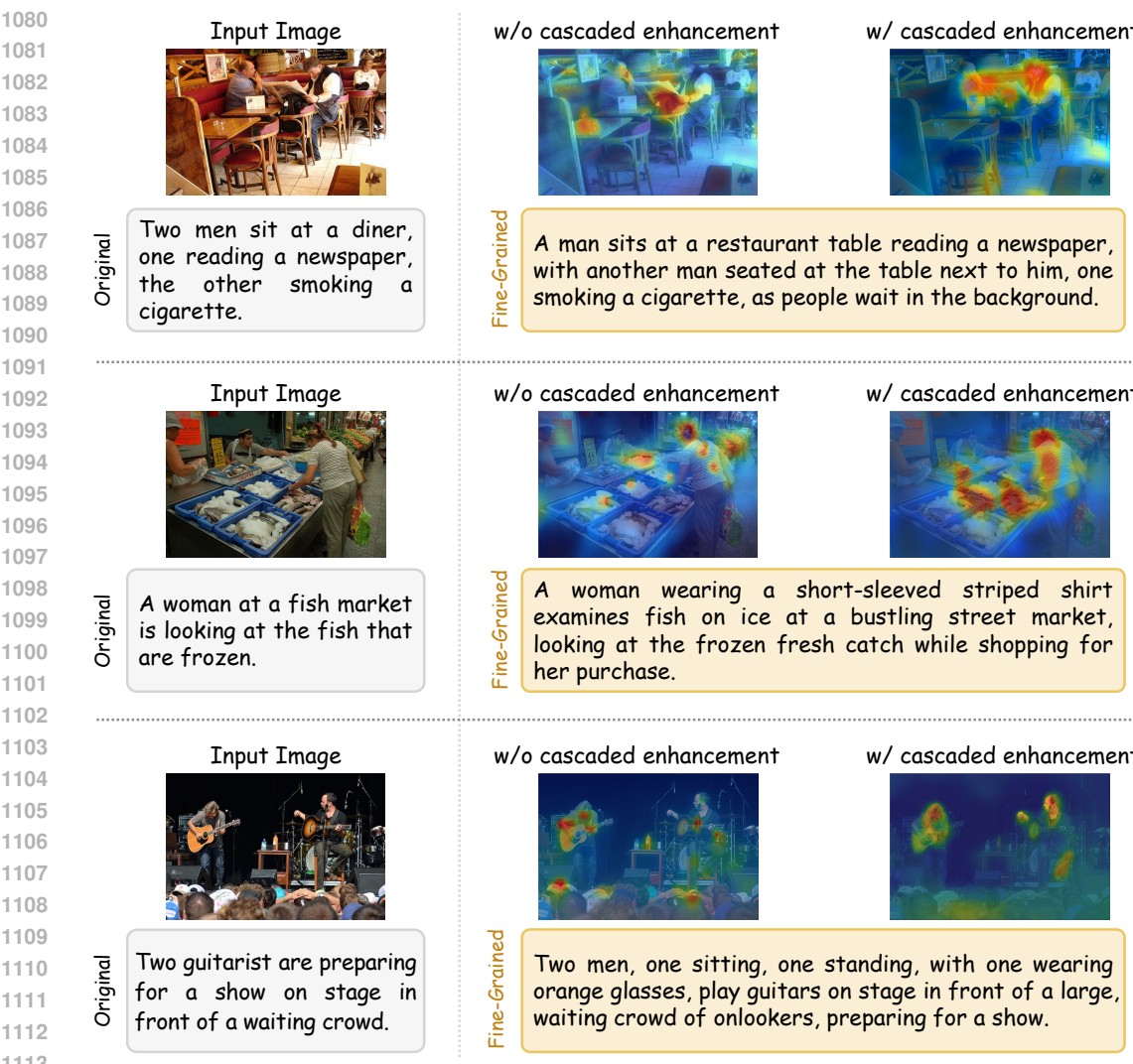

Figure 9: **Visual validation of Coarse-to-Fine Cascaded Enhancement (CE).** Our CE module addresses the context deficiency of fine-grained embeddings by injecting global contextual guidance. The comparison demonstrates that without CE (middle column), fine-grained attention struggles to model local information relationships. With CE (right column), our model achieves precise, contextually-aware alignment for complex fine-grained descriptions.

## B.9 VISUAL ANALYSIS OF CASCADED ENHANCEMENT

Fig. 9 provides a analysis to visually validate the efficacy of our Coarse-to-Fine Cascaded Enhancement (CE) module. The comparison demonstrates that without CE (middle column), fine-grained attention struggles with context deficiency, failing to accurately ground complex descriptions involving multiple entities or specific attributes. For instance, it cannot disambiguate the "man reading a newspaper" from the one "smoking a cigarette," nor can it precisely locate the "striped shirt" or the "orange glasses." Conversely, by injecting global contextual guidance, our CE module (right column) resolves these ambiguities, enabling precise, contextually-aware alignment. The resulting attention maps successfully disentangle parallel actions and ground fine-grained attributes to their corresponding image regions. This visual evidence substantiates our claim that CE is crucial for addressing the limitations of isolated fine-grained processing, enabling robust alignment for complex, multi-faceted image-text pairs.

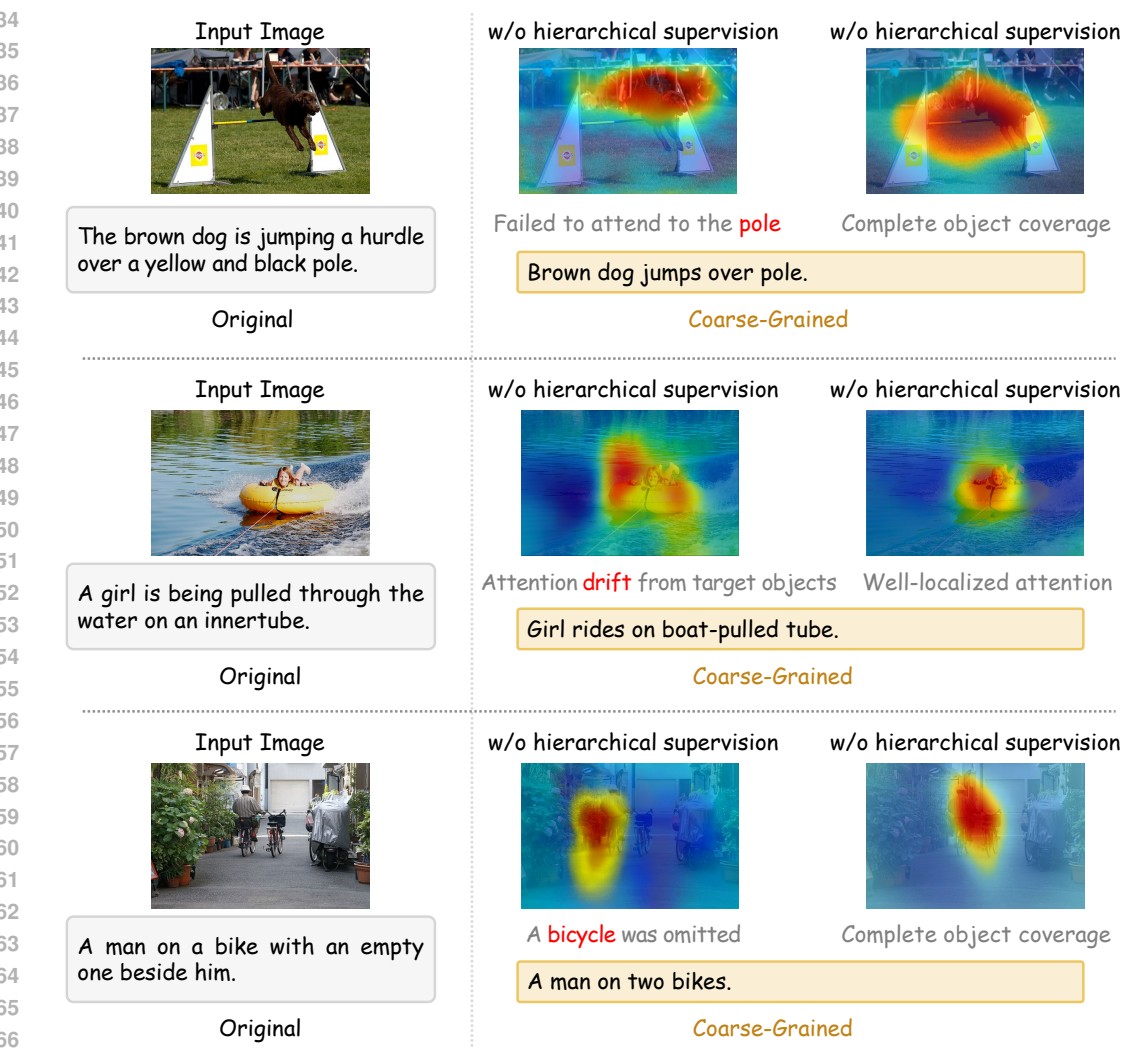

Figure 10: **Visual validation of Fine-to-Coarse Hierarchical Supervision (HS).** HS prevents semantic drift in coarse-grained representations. Without HS (middle), attention maps show common failures: missing key objects (second bicycle), poor component grounding (pole), or drift to irrelevant backgrounds. With HS (right), fine-level supervision guides coarse models to maintain semantic consistency, producing well-localized attention that accurately reflects textual descriptions.

## B.10 VISUAL ANALYSIS OF HIERARCHICAL SUPERVISION

Our Fine-to-Coarse Hierarchical Supervision (HS) plays a crucial role in mitigating semantic drift at coarser granularity levels, as visually validated in Fig. 10. Without HS, coarse-grained models trained on simplified text-image pairs often produce flawed alignments; the attention may drift to background noise (e.g., the water wake instead of the girl), omit less salient objects mentioned in the text (e.g., the second bicycle), or fail to ground all relevant components (e.g., ignoring the pole). These inconsistencies arise because coarser levels are optimized in isolation with ambiguous supervision. Our HS mechanism addresses this by using the finest-level alignment as a teacher distribution to regularize the learning process across the hierarchy. As demonstrated in the right column, this forces the coarse-grained representations to maintain semantic consistency, resulting in well-localized attention and complete object coverage that correctly reflects the underlying semantics.

## C    ETHICS STATEMENT

This work adheres to the ICLR Code of Ethics. Our research methodology does not involve human subjects or animal experimentation. All datasets utilized in this study are publicly available and were used in accordance with their respective licensing agreements and usage guidelines. We have ensured that no personally identifiable information was collected, processed, or disclosed during the course of this research. The experimental design and data processing procedures were developed to avoid potential biases and discriminatory outcomes.

## D    REPRODUCIBILITY STATEMENT

We are committed to ensuring the reproducibility of our work. Upon acceptance of this paper, we will make the source code publicly available. The experimental setup, including model configurations, training procedures, and key hyperparameters, are detailed throughout the paper. All datasets used in our experiments are either publicly available or will be released alongside the code to facilitate replication of our results.

## E    LLM USAGE

Large Language Models (LLMs) were used to assist in the writing and editing of this manuscript. Specifically, we employed LLMs for language polishing, improving readability, enhancing clarity, and assisting with various writing tasks throughout different sections of the paper. The models helped with sentence restructuring, grammar correction, style improvements, and ensuring coherent flow of the text.

All instances where LLMs contributed to the content generation or text refinement have been clearly identified and appropriately cited throughout the manuscript. We have maintained transparency by marking LLM-assisted sections and ensuring proper attribution where applicable.

It is important to emphasize that LLMs were not involved in the conceptualization, research methodology, experimental design, or data analysis phases of this work. All research ideas, hypotheses, methodological approaches, and scientific analyses were independently developed and conducted by the authors. The role of LLMs was strictly limited to improving the linguistic quality and presentation of the manuscript, without any contribution to the underlying scientific content, results interpretation, or conclusions.

The authors assume full responsibility for all content in this manuscript, including any text that was generated or refined with LLM assistance. We have carefully reviewed and validated all LLM-generated content to ensure it meets ethical standards and does not constitute plagiarism or scientific misconduct. All claims, arguments, and scientific contributions presented in this paper represent the original work and intellectual contributions of the authors.

