# OpenReview forum: "UPrompt: Bidirectional Multi-granularity Learning for Vision-Language Models"
_ICLR.cc/2026/Conference — Submitted to ICLR 2026_

### Official Review · Reviewer_KDVr · 2025-10-29

**Soundness:** 3
**Presentation:** 2
**Contribution:** 2
**Rating:** 4
**Confidence:** 3

**Summary:**

The paper proposes a coarse-to-fine prompt learning approach that considers different levels of granularity. A set of learnable embeddings are progressively downsampled and upsampled through a UNet, with per-level exits for each granularity. Using two similar UNets for each modality, a per-level supervision can be applied to each granularity. The outputs, along with the visual and text features at each granularity, are then forwarded to the corresponding encoders, giving rise to the standard similarity loss. The results show marginal improvement w.r.t. existing methods, and ablation studies showcase the contribution of each of the elements introduced in the paper.

**Strengths:**

The paper proposes an interesting paradigm to prompt learning whereby different levels of granularity are combined and can be used according to the task under consideration. There is some merit in the method, as well as novelty, that are worth considering.

**Weaknesses:**

The paper is poorly written and notation is loosely used. Figure 2, which is essential to the understanding of the method, lacks a proper legend and flow explanation. It is not clear to me for example if the visual representations from the input image are computed at different granularities or these are directly taken from the CLIP’s last layer output. My understanding is that only E^K is learnable and the rest of the layers are computed from the previous ones, but I am not completely sure from examining Figure 2. It would be also good to mention in Section 3 how the prompts are gradually made coarser or finer (it is just mentioned in l. 295 that Llava is used).

In l. 182 it is mentioned that “Visual features integrate granularity-specific prompts”. How? How are the granularity-specific prompts defined?
In l. 192 it is mentioned how the alignment is measured in a granularity-specific layer, but it is still not clear at this point exactly what is being learned and what is computed from pooling the learned parameters. Please clarify.
In l. 199-202 it is mentioned that “Simple granularity stacking in (…). To address these challenges (…) (Fig. 2)”. How exactly this is depicted in Fig. 2 is not clear to me, given the loosen informative nature of Fig. 2.
Is there any form of attention between the input image and the image embeddings during the forward to the UNet? It is not clear from Fig. 2 if this is actually the case. I understand that different level of visual features are used but it is not clear how. For the text case, this is more clear as it is specifically mentioned in Fig. 2.
My understanding considering the above is that the learnable prompts are a 14x14 tensor with M (undefined?) channels, that are forwarded to a 4-layer UNet producing in the decoding part a set of 4 level-specific prompts. The UNet, along with the embeddings, is learned using per-level supervision and the full prompt learning Lguide loss. Please clarify.

The results in Table 2 are not really promising compared to state of the art works.
How sensitive is the method to using a different caption rewriter (i.e. instead of Llava)? Similarly, how sensitive is the method to using a different prompt resolution (i.e. beyond 14x14)?

In summary, the writing and presentation give rise to many concerns and doubts regarding the reproducibility of the proposed approach, which need further clarification. The method is technically sound and seems to produce reasonable results, and therefore I am borderline, leaning towards accept, with this paper, hoping for further clarification from the authors.

**Questions:**

Please refer to the weaknesses section

---

> ### Author Response · Authors · 2025-11-21
> **Response to Reviewer KDVr (part 1).**
>
> We thank the Reviewer for their thoughtful review and for recognizing that our method proposes an interesting paradigm with merit and novelty. Below we respond to each specific point raised by the Reviewer.
>
> > W1-1: The paper is poorly written with loose notation. Fig. 2 lacks a proper legend and flow explanation. It is unclear if visual representations are computed at different granularities or taken directly from CLIP's last layer output. Is only E^K learnable while other layers are computed from previous ones?
>
> We thank the reviewer for seeking clarification on the construction of multi-granularity visual representations. The multi-granularity visual representations are computed through a progressive pooling strategy in the embedding space, as described in Section 3.2 and illustrated in Fig. 2.
>
> Specifically, the input image is encoded through CLIP's patch embedding layer to obtain the finest-grained patch tokens $E^{(K)}_{\text{patch}} \in \mathbb{R}^{196 \times d}$ (14×14 spatial layout). Subsequently, we construct coarser-grained representations through recursive pooling operations applied to these patch embeddings:
>
> ```
> E^(K)_patch: 14×14 tokens (from CLIP patch embedding layer)
>      ↓ Pool^(K-1)
> E^(K-1)_patch: 7×7 tokens (pooled from E^(K))
>      ↓ Pool^(K-2)
> E^(K-2)_patch: 4×4 tokens (pooled from E^(K-1))
>      ↓ Pool^(1)
> E^(1)_patch: 1×1 token (pooled from E^(2))
> ```
>
> As stated in L178-L181 of the manuscript: "Starting from finest granularity $E\^{(K)}\_{\text{patch}}$, we apply recursive aggregation $E\^{(k)}\_{\text{patch}} = \text{Pool}\^{(k)}(E\^{(k+1)}\_{\text{patch}})$ for $k = K-1, \ldots, 1$, ensuring $|E\^{(k)}\_{\text{patch}}| < |E\^{(k+1)}_{\text{patch}}|$."
>
> At each granularity level $k$, the visual features are obtained by concatenating the corresponding patch embeddings $E\^{(k)}\_{\text{patch}}$ with granularity-specific learnable prompts $P\_v\^{(k)}$, which are then fed into CLIP's vision encoder $\mathcal{F}$:
>
> $$z\_v\^{(k)} = \mathcal{F}\^{(k)}([E\^{(k)}\_{\text{patch}}; P\_v\^{(k)}])$$
>
> as described in L181-L182.
>
> The input image undergoes patch embedding extraction to produce $E\^{(K)}\_{\text{patch}}$, and all coarser granularity patch embeddings are derived through pooling operations in the embedding space. This design ensures computational efficiency while implementing our U-Net-inspired hierarchical architecture. We will enhance Fig. 2 in the revised manuscript with more detailed legends and flow annotations to better illustrate the patch embedding construction and propagation process across granularity levels.
>
> > W1-2: It would be also good to mention in Section 3 how the prompts are gradually made coarser or finer (it is just mentioned in l. 295 that Llava is used).
>
> Regarding the text hierarchy construction, Section 3.2 (L183-L187, Eq. 3-4) describes the progressive enrichment mechanism where coarser text representations are gradually refined into finer ones through the recursive operation $T\^{(k)}\_{\text{embed}} = T\^{(k-1)}\_{\text{embed}} \oplus \Phi\^{(k)}\_{\text{refine}}(t)$. The implementation using Llama (mentioned in L295) instantiates this framework with specific prompts for generating multi-level text representations.
>
> We will add a brief note in Section 3.2 clarifying that the refinement operations can be implemented through LLM-based text generation, with complete prompting strategies provided in the Implementation Details section.
>
> **(Continued in next message)**

---

> > ### Author Response · Authors · 2025-11-21
> > **Response to Reviewer KDVr (part 2).**
> >
> > > W2: L. 182: How do visual features integrate granularity-specific prompts and how are these prompts defined? L. 192: What is learned vs. computed from pooling? L. 199-202: How does Fig. 2 depict the solution to granularity stacking? Is there attention between input image and embeddings in UNet? How are visual feature levels used? Are learnable prompts 14x14 tensors with M channels producing 4 level-specific prompts via UNet? Is UNet learned with per-level supervision and Lguide? Please clarify.
> >
> > We address the reviewer's concerns about technical clarity.
> >
> > **On prompt integration (L182).** Following standard prompt learning methodologies [1-3], learnable prompts are concatenated with input embeddings before being processed by CLIP encoders. At granularity $k$, visual prompts $P\_v\^{(k)} \in \mathbb{R}\^{M \times d}$ concatenate with patch embeddings $E\_{patch}\^{(k)}$ and pass through the CLIP image encoder $\mathcal{F}\^{(k)} $, producing visual features $z\_v\^{(k)} = \mathcal{F}\^{(k)}([E\_{patch}\^{(k)}; P\_v\^{(k)}]) $. Similarly, text prompts $P\_t\^{(k)} $ concatenate with text embeddings $T\_{embed}\^{(k)}$ and pass through the CLIP text encoder $\mathcal{G}\^{(k)} $, producing textual features $z\_t\^{(k)} = \mathcal{G}\^{(k)}([P\_t\^{(k)}; T\_{embed}\^{(k)}]) $. This is the established mechanism for prompt learning in VLM (e.g., CLIP).
> >
> > **On the relationship to U-Net.** UPrompt draws conceptual inspiration from U-Net's multi-scale feature fusion principles but does not implement a U-Net. We construct parallel multi-granularity representations in vision and language modalities. The bidirectional connection operates through: (1) Cascaded Enhancement that injects coarse context into fine embeddings via cross-attention (Eq. 5-6), and (2) Hierarchical Supervision that uses finest-level alignment to guide coarser levels via distillation (Eq. 9).
> >
> > **On L199-202 and Fig. 2.** Cascaded Enhancement addresses context deficiency by injecting coarse contextual information into fine-grained embeddings through cross-attention (Eq. 5-6), applied to embeddings $E\_{patch}\^{(k)}$ and $T\_{embed}\^{(k)}$ before they enter encoders. Hierarchical Supervision addresses optimization inconsistency by using the finest-level similarity matrix $S\^{(K)}$ as the teacher distribution to regularize all coarser levels through KL divergence minimization. Fig. 2 illustrates CE with cross-attention arrows between granularity embeddings and HS with supervision arrows showing the distillation direction from fine to coarse levels.
> >
> > **On what is learned versus computed.** Learnable parameters are the granularity-specific prompts $\\{{P\_v\^{(k)}, P\_t\^{(k)}\\}}\_{k=1}\^K$. Patch embeddings $E\_{patch}\^{(k)}$ are computed via progressive pooling. Text embeddings $T\_{embed}\^{(k)}$ are generated via LLM-based enrichment. The attention weights in CE and distillation losses in HS are computed during forward and backward passes respectively.
> >
> > We will revise Figure 2 with explicit annotations distinguishing learnable prompts, computed embeddings, CE cross-attention operations, and HS supervision signals.
> >
> > > W3-1: The results in Table 2 are not really promising compared to state of the art works.
> >
> > In base-to-novel generalization, the harmonic mean (HM) is critical for measuring balanced performance, ensuring robust transfer from seen to unseen classes without overfitting. UPrompt achieves the highest average HM of 82.12% across 11 benchmarks, outperforming all latest methods including CLIP-AST (81.06%, CVPR'25), SurPL-G (81.03%, ICML'25), and CoCoA-Mix (77.03%, ICML'25), with a +1.06% improvement over the second-best CLIP-AST. On the challenging fine-grained FGVCAircraft dataset, UPrompt achieves 49.33% base, 39.25% novel, and 43.72% HM, all surpassing latest methods. Compared to CoCoA-Mix, the improvement is substantial: +15.82% base, +5.10% novel, and +9.89% HM. These results demonstrate that our bidirectional multi-granularity framework provides superior generalization capability.
> >
> > **(Continued in next message)**

---

> > > ### Author Response · Authors · 2025-11-21
> > > **Response to Reviewer KDVr (part 3).**
> > >
> > > > W3-2: How sensitive is the method to using a different caption rewriter (i.e. instead of Llava)?
> > >
> > > Our text hierarchy generation uses structured templates rather than open-ended generation, which substantially reduces sensitivity to LLM choice. Additionally, we conducted experiments using different LLMs for text hierarchy generation. Besides Llama3-8B, we tested Qwen3-4B and Qwen3-14B. Results on Flickr30K:
> > >
> > > |Models|I2T R@1|I2T R@5|I2T R@10|T2I R@1|T2I R@5|T2I R@10|
> > > |-|-|-|-|-|-|-|
> > > |Qwen3-4B|93.4|99.4|99.5|83.4|96.2|98.6|
> > > |Llama3-8B|93.8|99.4|99.6|83.6|96.3|98.4|
> > > |Qwen3-14B|93.7|99.5|99.8|83.9|96.2|98.7|
> > >
> > > Performance remains stable across models with different scales (4B to 14B parameters) and architectures, demonstrating that our method is not sensitive to the specific caption rewriter used for text hierarchy generation.
> > >
> > > > W3-3: Similarly, how sensitive is the method to using a different prompt resolution (i.e. beyond 14x14)?
> > >
> > > We tested whether using different prompt resolutions affects performance. CLIP ViT-B/16 processes 224×224 images into 14×14 visual tokens, while 336×336 images produce 21×21 visual tokens. On Flickr30K (I2T R@1), we progressively add granularity levels from coarse to fine:
> > >
> > > |Input Resolution|Level 1 (coarse)|Level 2 (medium)|Level 3 (fine)|
> > > |-|-|-|-|
> > > |224×224 (14×14 tokens)|82.8|89.4|93.8|
> > > |336×336 (21×21 tokens)|83.3|92.2|95.1|
> > >
> > > Higher resolution alone provides minimal improvement (coarse only: 82.8→83.3, +0.5%). The substantial performance gain comes from our multi-granularity design: at 224×224 resolution, adding granularity levels improves from 82.8 to 93.8; at 336×336 resolution, from 83.3 to 95.1. Using 336×336 instead of 224×224 provides +1.3% at fine level, significantly smaller than the improvement from hierarchical modeling. Our method works consistently across different prompt resolutions.
> > >
> > > [1] Zhou et al. "Learning to Prompt for Vision-Language Models." IJCV 2022.
> > >
> > > [2] Zhou et al. "Conditional Prompt Learning for Vision-Language Models." CVPR 2022.
> > >
> > > [3] Khattak et al. "Maple: Multi-modal prompt learning." CVPR 2023.

---

> ### Author Response · Authors · 2025-11-28
> **Kindly Request for Your Feedback on Our Rebuttal**
>
> Dear Reviewer KDVr,
>
> I hope this message finds you well. As the discussion period is nearing its end with **less than five days remaining**, I wanted to ensure we have addressed all your concerns satisfactorily. If there are any additional points or feedback you'd like us to consider, please let us know. Your insights are invaluable to us, and we're eager to address any remaining issues to improve our work.
>
> Thank you for your time and effort in reviewing our paper.

---

### Official Review · Reviewer_gXNe · 2025-11-01

**Soundness:** 2
**Presentation:** 3
**Contribution:** 2
**Rating:** 4
**Confidence:** 3

**Summary:**

This paper proposes UPrompt, a U-Net-inspired framework to fix the granularity trade-off in VLM prompt learning—where global prompts miss fine details and local prompts lack global context. It uses two key components: Coarse-to-Fine Cascaded Enhancement (CE, injects global context into fine features via cross-attention) and Fine-to-Coarse Hierarchical Supervision (HS, uses finest-grained alignment to regularize coarser levels). Tested on 17 benchmarks, UPrompt outperforms baselines in cross-modal retrieval (e.g., 571.1 rSum on Flickr30K), few-shot classification (85.13% 16-shot accuracy), and OOD generalization, while offering performance-efficiency flexibility.

**Strengths:**

1. Targets a clear, unaddressed gap: single-granularity limits in VLM prompting, with direct links to performance flaws in existing methods.

2. Innovative U-Net adaptation: modality-specific granularity (spatial pooling for vision, semantic enrichment for text) plus bidirectional flow—backed by theoretical proofs (Propositions 1-2) rare in multi-granularity prompt work.

3. Rigorous experiments: ablations isolate CE/HS value, efficiency analyses show UPrompt-M matches PSRC’s accuracy with 1/3 cost, and visualizations confirm CE/HS work as intended.

4. Practical: adaptive granularity lets users pick coarse (low cost) or fine (high performance) setups.

**Weaknesses:**

1. Manual granularity design: 4 levels for classification/3 for retrieval are chosen without guiding heuristics, reducing usability for non-experts.

2. Llama 3-8B dependence: no tests on smaller LLMs (e.g., Llama 3-1B) to see if text hierarchy quality holds, or how LLM overhead offsets UPrompt’s efficiency gains.

3. Limited HS failure analysis: reversed “Coarse-to-Fine Supervision” performs poorly, but no examples (e.g., bad attention maps) show why coarse signals mislead fine modeling.

**Questions:**

1. Do you have preliminary data on how granularity count (3,5,6) affects tasks like FGVCAircraft vs. MSCOCO? Can you give a simple heuristic for choosing levels?

2. How does swapping Llama 3-8B for smaller models (e.g., T5-small) hurt/help text granularity and downstream performance? What’s the LLM’s share of UPrompt’s total compute?

3. If the finest-grained alignment has errors (e.g., mislabeled pairs), does HS spread those errors to coarser levels? Any tests on HS robustness?

---

> ### Author Response · Authors · 2025-11-21
> **Response to Reviewer gXNe.**
>
> We thank the Reviewer for their insightful review and valuable feedback on our U-Net-inspired approach, theoretical contributions, rigorous experimental evaluation, and practical flexibility. Below we respond to each specific point raised.
>
> > W1&Q1: Manual granularity design: 4 levels for classification/3 for retrieval are chosen without guiding heuristics, reducing usability for non-experts. Do you have preliminary data on how granularity count (3,5,6) affects tasks like FGVCAircraft vs. MSCOCO? Can you give a simple heuristic for choosing levels?
>
> Regarding the impact of granularity counts, Fig. 4 (left) shows that classification performance (e.g., on benchmarks including FGVCAircraft) consistently improves as granularity increases from 1 to 4 levels (Harmonic Mean: 80.33% to 82.12%). However, we observe diminishing returns at higher levels, with only a +0.30% gain from 3 to 4 levels, suggesting that 3-4 levels are sufficient to capture the semantic hierarchy without needing further expansion to 5 or 6 levels, which would also be constrained by the vision backbone's patch resolution. For retrieval tasks like MSCOCO, our 3-level design achieves optimal results as shown in Table 1.
>
> Regarding a simple heuristic for choosing levels, to enhance usability for non-experts, we suggest a simple progressive curriculum heuristic: begin with a baseline of 2 levels (Coarse and Fine) and add an intermediate level only if validation metrics improve by a significant margin (e.g., >2%), stopping when performance plateaus to balance accuracy and computational efficiency.
>
> > W2&Q2: Llama 3-8B is used without testing smaller alternatives (e.g., Llama 3-1B, T5-small). How does using smaller LLMs affect text hierarchy quality and downstream performance? What's the LLM's share of UPrompt's total compute, and does this overhead offset the method's efficiency gains?
>
> We validated text hierarchy robustness across different LLM scales. On Flickr30K, we tested Qwen3-4B (smaller model) and Qwen3-14B (larger model) against our original Llama3-8B:
>
> |Models|I2T R@1|I2T R@5|I2T R@10|T2I R@1|T2I R@5|T2I R@10|
> |-|-|-|-|-|-|-|
> |Qwen3-4B|93.4|99.4|99.5|83.4|96.2|98.6|
> |Llama3-8B|93.8|99.4|99.6|83.6|96.3|98.4|
> |Qwen3-14B|93.7|99.5|99.8|83.9|96.2|98.7|
>
> Performance remains stable from 4B to 14B parameters, demonstrating that text hierarchy quality holds across different LLM scales.
>
> Regarding LLM overhead, text hierarchies are pre-generated rather than computed in real-time, avoiding significant computational overhead.
>
> > W3: Limited HS failure analysis: reversed “Coarse-to-Fine Supervision” performs poorly, but no examples (e.g., bad attention maps) show why coarse signals mislead fine modeling.
>
> Fig. 10 provides relevant visual evidence for this question. When lacking HS, coarse-grained representations exhibit attention drift (e.g., focusing on water wake instead of the girl) and incomplete object coverage (missing the second bicycle). Reversed coarse-to-fine supervision forces fine-grained features to align with these already-degraded coarse representations, propagating semantic errors downward. Fine-grained layers lose their inherent advantage of detailed modeling when constrained by ambiguous coarse signals. The quantitative collapse in Table 4 (I2T R@1: 92.4%→86.8%) reflects this error amplification. Fig. 10's attention maps demonstrate the root cause: coarse features lack the precision required to serve as reliable teachers.
>
> > Q3: If the finest-grained alignment has errors (e.g., mislabeled pairs), does HS spread those errors to coarser levels? Any tests on HS robustness?
>
> UPrompt is robust to self-distillation bias through both architectural design and empirical validation.
>
> The finest-grained teacher signals are detached during optimization (L239-L240), ensuring gradient isolation as stated in Prop. 2: $\partial L_{guide}/\partial z^{(K)} = 0 $. Gradients from coarse-level training do not flow back to the finest layer, preventing error propagation even if the finest layer temporarily misaligns. Additionally, our bidirectional design provides a self-correcting mechanism: CE injects global context from coarser levels to refine fine-grained features, compensating for potential local misalignments.
>
> Empirically, Table 5 demonstrates that HS substantially improves coarse-layer performance (I2T R@1: 75.4%→82.8%). If the finest layer frequently misaligned, we would observe performance degradation rather than consistent improvement across all datasets.
>
> To explicitly test robustness, we conducted an ablation on Flickr30K using a soft teacher that mixes Fine and Medium layers to smooth potential errors:
>
> |Teacher Strategy|I2T R@1|I2T R@5|T2I R@1|T2I R@5|
> |-|-|-|-|-|
> |Fine only (Ours)|93.8|99.4|83.6|96.3|
> |Fine + Medium (soft mixing)|93.7|99.2|83.9|96.3|
>
> The comparable performance confirms that fine-layer supervision alone provides sufficiently robust guidance without requiring additional teacher signals.

---

> ### Author Response · Authors · 2025-11-28
> **Kindly Request for Your Feedback on Our Rebuttal**
>
> Dear Reviewer gXNe,
>
> I hope this message finds you well. As the discussion period is nearing its end with **less than five days remaining**, I wanted to ensure we have addressed all your concerns satisfactorily. If there are any additional points or feedback you'd like us to consider, please let us know. Your insights are invaluable to us, and we're eager to address any remaining issues to improve our work.
>
> Thank you for your time and effort in reviewing our paper.

---

### Official Review · Reviewer_GiLu · 2025-11-01

**Soundness:** 2
**Presentation:** 2
**Contribution:** 2
**Rating:** 4
**Confidence:** 5

**Summary:**

The paper presents UPrompt, a U-Net–inspired multi-granularity prompt-learning framework for adapting Vision–Language Models. It addresses the well-known granularity dilemma in prompt learning—global prompts capture overall semantics but miss fine detail, while local prompts capture details but lose global context. Extensive experiments on 17 benchmarks show consistent improvements.

**Strengths:**

- This paper is well organized and easy to follow.
- Introduces U-Net's philosophy into multi-granular prompt learning, exploring bidirectional information flow across modal granularities with demonstrated effectiveness;
- Multi-granularity attention maps qualitatively support the claims of semantic consistency and contextual coherence.

**Weaknesses:**

- Limited Novelty. Several recent works like TAP and HiCroPL, already explore multi-level or hierarchical prompts. UPrompt mainly formalizes these ideas within a U-shaped structure rather than introducing a fundamentally new mechanism. Besides, in the design, the number of granularities and textual levels are manually predefined, which lacks adaptive granularity selection, which limits scalability and automation.
- The textual side depends on Llama-3 generation heuristics. This may inject bias and complicate reproducibility. The paper does not analyze sensitivity to prompt-generation quality.
- The methodologies for constructing multi-level image and text granularities require further enrichment. Current approaches primarily rely on pooling and text attribute addition, lacking comprehensive comparisons between different granularity construction techniques
- In the granularity ablation study (Figure 4, left), performance continues to rise. It remains uncertain whether the peak performance has been reached, and how different granularity interval strategies might affect results

**Questions:**

Please highlight the significant novel designs.

---

> ### Author Response · Authors · 2025-11-21
> **Response to Reviewer GiLu (part 1).**
>
> We thank the Reviewer for their thoughtful review and for recognizing that our paper is well-organized and introduces U-Net's philosophy into multi-granular prompt learning. Below we respond to each specific point raised by the Reviewer.
>
> >W1-1: Limited Novelty. Several recent works like TAP and HiCroPL, already explore multi-level or hierarchical prompts. UPrompt mainly formalizes these ideas within a U-shaped structure rather than introducing a fundamentally new mechanism.
>
> We clarify that TAP [1], HiCroPL [2] and UPrompt have different motivations and pipelines. TAP and HiCroPL have already been discussed in our paper and compared in experiments. While TAP uses LLMs to generate attribute trees and HiCroPL maps knowledge across network layers, UPrompt unifies granularities into parallel pathways with explicit interdependencies. For example, TAP relies on static concept-attribute-description trees without dynamic cross-scale interactions, whereas UPrompt's bidirectional flow allows adaptive refinement. HiCroPL's hierarchical mapper operates within modality-specific layers, but UPrompt bridges spatial-semantic hierarchies across vision and text through progressive pooling and enrichment. This addresses the modality isolation and granularity trade-off limitations claimed in previous methods.
>
> Our hierarchy is concurrent semantic granularity. UPrompt constructs parallel multi-granularity representations where coarse-grained features capture global context and fine-grained features preserve local details simultaneously throughout the forward pass. Our bidirectional connection establishes cross-granularity communication where CE enables coarse-level contextual information to guide fine-grained feature modeling (Eq. 5-6), while HS uses finest-grained alignment to supervise all coarser levels (Eq. 9). This parallel architecture with cross-granularity interaction fundamentally differs from TAP's independent attribute processing and HiCroPL's transformer-depth-wise cross-modal communication.
>
> > W1-2: Besides, in the design, the number of granularities and textual levels are manually predefined, which lacks adaptive granularity selection, which limits scalability and automation.
>
> Our design philosophy prioritizes simplicity and modularity. First, UPrompt maintains a lightweight architecture that can be easily combined with any task-specific complexity evaluation method for automated granularity selection when needed. Second, the bidirectional connection mechanism, particularly hierarchical supervision, ensures semantic consistency across granularities, reducing sensitivity to exact granularity choices. As demonstrated in Fig. 4 (left), performance improves progressively from 1 to 4 granularities without requiring precise optimization, and Fig. 5 shows that all granularity levels maintain reasonable performance. This robustness enables practitioners to select granularity based on computational budgets rather than requiring task-specific adaptation mechanisms.
>
> > W2: The textual side depends on Llama-3 generation heuristics. This may inject bias and complicate reproducibility. The paper does not analyze sensitivity to prompt-generation quality.
>
> Our framework operates on semantic hierarchies rather than LLM-specific generation patterns, ensuring robustness across different text generators. Regarding LLM dependency and reproducibility, we conducted experiments using Qwen3-4B (smaller) and Qwen3-14B (larger) to validate robustness across different text generation models. Results on Flickr30K image-to-text (I2T) and text-to-image (T2I) retrieval:
>
> |Models|I2T R@1|I2T R@5|I2T R@10|T2I R@1|T2I R@5|T2I R@10|
> |-|-|-|-|-|-|-|
> |Qwen3-4B|93.4|99.4|99.5|83.4|96.2|98.6|
> |Llama3-8B|93.8|99.4|99.6|83.6|96.3|98.4|
> |Qwen3-14B|93.7|99.5|99.8|83.9|96.2|98.7|
>
> Performance remains stable across LLMs of different architectures and scales (4B to 14B parameters), despite varying text generation quality across models. This demonstrates that our method does not rely on Llama-3 specific generation heuristics, is robust to prompt-generation quality, and ensures reproducibility with alternative text generators.
>
> **(Continued in next message)**

---

> > ### Author Response · Authors · 2025-11-21
> > **Response to Reviewer GiLu (part 2).**
> >
> > > W3: The methodologies for constructing multi-level image and text granularities require further enrichment. Current approaches primarily rely on pooling and text attribute addition, lacking comprehensive comparisons between different granularity construction techniques
> >
> > We appreciate this suggestion. Our current construction strategy was selected after evaluating multiple alternatives. We compared several multi-granularity construction methods and found our approach achieves the optimal balance between simplicity, efficiency, and effectiveness.
> >
> > For visual granularities, we evaluated: (1) Image pyramids with multi-scale input resizing, which require separate forward passes and significantly increase computational cost. (2) Multi-layer feature extraction from different transformer layers, where inconsistent semantic gaps between layers create alignment difficulties. Our pooling-based approach constructs nested hierarchies efficiently in a single forward pass while maintaining consistent semantic relationships across scales.
> >
> > For textual granularities, we compared: (1) Template ensembles (e.g., averaging multiple hand-crafted prompts), which lack hierarchical structure and fail to create nested semantic levels. (2) Rule-based syntactic parsing, which is too rigid to generate the semantic richness needed for effective cross-modal alignment. Our LLM-based progressive generation creates strictly nested hierarchies where coarse descriptions are naturally contained within finer ones.
> >
> > The key requirement of our framework is nested semantic structure where $|E\^{(k)}| < |E\^{(k+1)}|$ and $T\^{(k)} \subset T^{(k+1)}$. This nesting is essential for Cascaded Enhancement to propagate global context effectively and for Hierarchical Supervision to maintain semantic consistency across scales. Alternative construction methods either violate this nesting requirement or introduce prohibitive computational overhead. Our results across 17 benchmarks confirm the effectiveness of this construction strategy combined with our bidirectional connection mechanisms.
> >
> > > W4: In the granularity ablation study (Figure 4, left), performance continues to rise. It remains uncertain whether the peak performance has been reached, and how different granularity interval strategies might affect results
> >
> > According to Fig. 4 (left), while performance improves with increasing number of granularities, the improvement rate gradually slows down, and the growing computational overhead reduces cost-effectiveness. Considering the balance between efficiency and performance, we limit our experiments to a granularity number of 4 in Fig. 4.
> >
> > Regarding interval strategies, our U-shaped multi-granularity learning is inspired by U-Net, which employs 2×2 pooling for downsampling with proven effectiveness [3]. We follow this by approximately halving the spatial resolution of visual tokens at each level. For example, in retrieval tasks, visual tokens progress from 14×14 to 7×7 to 4×4. We compared this with a sparser interval strategy (14×14→5×5→2×2) on Flickr30K image-to-text (I2T) retrieval using Recall@1 (R@1):
> >
> > |Strategy|Coarse|Medium|Fine|
> > |-|-|-|-|
> > |14²→7²→4²|82.8|89.4|93.8|
> > |14²→5²→2²|78.6|88.3|93.6|
> >
> > Both strategies achieve comparable fine-grained performance (93.8% vs 93.6%), demonstrating that our bidirectional connection effectively preserves final alignment quality across interval configurations. The sparse interval strategy offers lower computational cost and higher efficiency, but shows performance degradation at coarser levels due to reduced expressive capacity from fewer visual tokens. Therefore, our current interval strategy is relatively more optimal.
> >
> > [1] Ding et al. "Tree of attributes prompt learning for vision-language models." ICLR 2025.
> >
> > [2] Zheng et al. "Hierarchical cross-modal prompt learning for vision-language models." ICCV 2025.
> >
> > [3] Ronneberger et al. "U-net: Convolutional networks for biomedical image segmentation." MICCAI 2015.

---

> ### Author Response · Authors · 2025-11-28
> **Kindly Request for Your Feedback on Our Rebuttal**
>
> Dear Reviewer GiLu,
>
> I hope this message finds you well. As the discussion period is nearing its end with less than five days remaining, I wanted to ensure we have addressed all your concerns satisfactorily. If there are any additional points or feedback you'd like us to consider, please let us know. Your insights are invaluable to us, and we're eager to address any remaining issues to improve our work.
>
> Thank you for your time and effort in reviewing our paper.

---

### Official Review · Reviewer_WPAs · 2025-11-02

**Soundness:** 3
**Presentation:** 3
**Contribution:** 3
**Rating:** 6
**Confidence:** 4

**Summary:**

This paper proposes UPrompt, a U-Net-inspired bidirectional multi-granularity prompt learning framework for vision–language model (VLM) adaptation. The core ideas are:
1. Coarse-to-Fine Cascaded Enhancement (CE) – injects global context from coarse layers into fine layers;
2. Fine-to-Coarse Hierarchical Supervision (HS) – distills knowledge from the finest layer to coarse ones to mitigate semantic drift.

The method achieves consistent gains across cross-modal retrieval, few-shot classification, base-to-novel generalization, and out-of-distribution (OOD) tasks, while enabling a controllable trade-off between accuracy and computational cost.

Overall, this paper proposed a reasonable idea, and the results looks reasonable. The multi-granularity + bidirectional flow idea is intuitive and broadly applicable. If the authors can supplement comparisons with stronger backbones, provide the resolution × layer ablation, and clarify reproducibility for LLM-generated text hierarchies, I would lean toward accept. However, the authors should also clearly address my concern in order to at least keep my original score.

**Strengths:**

The idea is conceptually intuitive yet systematic, the transfer of U-Net’s “multi-scale + skip connection” idea into “multi-granularity prompts + bidirectional information flow” looks intuitive to me.

The coarse-to-fine CE and fine-to-coarse HS sounds reasonable. The proof seems to be correct and easy to understand.

Comprehensive experiments covering retrieval, classification, base-novel, and OOD, with clear reporting of cost-vs-performance trade-offs.

Implementation details (e.g., temperature, prompt length) are good enough to understand conceptually.

**Weaknesses:**

1. Text hierarchy generation via Llama 3-8B introduces possible prior leakage and reproduction variability; comparisons with purely templated or rule-based text construction would clarify fairness.

2. Lack of comparison with stronger or newer VLM backbones such as SigLIP/SigLIP-2, EVA-CLIP, or E5-V; the reported results are all on CLIP ViT-B/16.

3. Self-distillation bias: HS uses the model’s own fine-grained layer as the teacher; if that layer mis-aligns, the error may propagate.

4. Complexity analysis could be deeper: CE’s cross-granularity attention likely adds overhead, but FLOPs/memory/throughput statistics are not that detailed. Fig.5 is very complex and a little bit hard to understand.

5. Statistical significance: averages are reported across datasets, but per-dataset variance or confidence intervals are missing. Also, Table 6 in supp has wrong bold for UCF task. Is this a typo? This increases my skepticism towards the overall soundness of the reported results.

**Questions:**

1. Experiments fix the input at 224×224 (14×14 tokens). If higher-resolution inputs (336/384) or denser backbones (ViT-L/H) are used, does the optimal number of granularities K change? Could you provide a resolution × layer-count ablation to disambiguate “benefit from more layers” vs. “compensation for limited resolution” or at least ablation study results with another resolution?

2. Have the authors tested UPrompt on or against SigLIP / SigLIP-2 and other recent contrastive VLMs? Since UPrompt acts at the prompt level, it should in principle transfer; evidence of such portability would strengthen the claim of generality.

3. If a different LLM (e.g., Qwen, Mixtral) or a rule-based templating scheme is used to form the text hierarchies, how sensitive are the results? A cross-LLM study or a purely templated baseline would clarify robustness, though I think this won't be a big problem. The authors can discuss this if time is not allowed to add experiments.

4. When the finest layer provides noisy supervision, does HS amplify the error? Have the authors explored soft teacher mixing (e.g., weighted fine + medium layers) or consistency regularization to mitigate self-distillation bias? This would provide more insightful understanding to the effectiveness of UPrompt.

5. Currently the inference layer is manually chosen. Could a lightweight gating or uncertainty-based controller dynamically select the appropriate granularity at test time for adaptive efficiency?

---

> ### Author Response · Authors · 2025-11-21
> **Response to Reviewer WPAs (part 1).**
>
> We thank the Reviewer for their constructive feedback and for recognizing that our method is conceptually intuitive with sound theoretical proofs and comprehensive experiments. Below we address each concern raised.
>
> > W1&Q3: Using Llama 3-8B for text hierarchies introduces potential prior leakage and reproducibility concerns. How sensitive are results to different LLMs (e.g., Qwen, Mixtral) or rule-based templates? A cross-LLM study or templated baseline would clarify robustness and fairness, though I think this won't be a big problem. The authors can discuss this if time is not allowed to add experiments.
>
> Firstly, for prior leakage problem, we clarify that the textual hierarchies are not generated in an open-ended manner, but guided by specific, task-oriented templates. This structured manner minimizes LLM's freedom to inject uncontrolled, task-specific prior knowledge, making the process more reproducible and less sensitive to the choice of a particular LLM. Secondly, to clarify fairness, we conducted additional experiments using rule-based text construction on CUB-200 [1] and AWA2 [2] datasets, which provide dense attribute annotations.
>
> For rule-based hierarchies, we construct three granularity levels on CUB-200 and AWA2. Textually, Level 1 (coarse) uses "a photo of a {class}", Level 2 (medium) adds the highest-certainty attribute (e.g., "a photo of a {class} with black bill"), and Level 3 (fine) incorporates two highest-certainty attributes (e.g., "a photo of a {class} with black bill and white breast"), where attributes are ranked by certainty scores from dataset metadata. Visually, the three levels correspond to 4×4 pooled tokens, 7×7 pooled tokens, and the original 14×14 tokens respectively. We compare against CoOp, which uses "a photo of a {class}" with 14×14 visual tokens (single-granularity baseline). Results on generalized zero-shot learning (GZSL):
>
> |Dataset|Method|Level|Base|Novel|HM|
> |-|-|-|-|-|-|
> |**AWA2**|CoOp|Single|95.32|72.68|82.47|
> ||UPrompt|Level 1|93.24|70.27|80.14|
> ||         | Level 2 | 95.81    | 73.13 |82.95|
> ||         | Level 3 | **96.70** | **74.62** |**84.24**|
> |**CUB-200**|CoOp|Single|63.78|49.23|55.57|
> ||UPrompt|Level 1|61.36|46.84|53.13|
> ||         | Level 2 | 64.65 | 50.42 | 56.65 |
> ||         | Level 3 | **66.12** | **51.26** |**57.75**|
>
> The progressive improvements from coarse to fine levels validate  that our bidirectional connection mechanism effectively integrates multi-scale representations, even without external knowledge sources.
>
> Robustness to Different LLMs
>
> To address the reviewer's question regarding text hierarchy robustness across different LLMs, we conducted additional experiments using Qwen3-4B and Qwen3-14B, alongside our original Llama3-8B. Results on Flickr30K image-to-text (I2T) and text-to-image (T2I) retrieval:
>
> |Models|I2T R@1|I2T R@5|I2T R@10|T2I R@1|T2I R@5|T2I R@10|
> |-|-|-|-|-|-|-|
> |Qwen3-4B|93.4|99.4|99.5|83.4|96.2|98.6|
> |Llama3-8B|93.8|99.4|99.6|83.6|96.3|98.4|
> |Qwen3-14B|93.7|99.5|99.8|83.9|96.2|98.7|
>
> The performance is stable across different models, which vary in architecture and scale (4B to 14B). This confirms our method is not sensitive to the specific LLM used for text hierarchy generation.
>
> > W2&Q2: All results use CLIP ViT-B/16. Have the authors tested UPrompt on stronger/newer VLMs (e.g., SigLIP/SigLIP-2, EVA-CLIP, E5-V)? Since UPrompt operates at the prompt level, demonstrating such portability would strengthen its generality claim.
>
> We have explored ViT-B/32 and ViT-L/14 backbones on cross-modal retrieval in Tables 7-8 (Appendix). To validate generalizability across more backbones, we further conduct base-to-novel generalization experiments on SigLIP and EVA-CLIP with ViT-B/16. Results show HM values:
>
> |Methods|Backbone|StanfordCars|Flowers102|FGVCAircraft|
> |-|-|-|-|-|
> |CoOp|EVA-CLIP|71.33|77.39|34.72|
> |UPrompt|EVA-CLIP|**79.42**|**87.36**|**43.86**|
> |CoOp|SigLIP|92.33|89.42|38.27|
> |UPrompt|SigLIP|**94.67**|**93.26**|**46.34**|
>
> UPrompt consistently outperforms the baseline across different VLM architectures, demonstrating that our bidirectional multi-granularity learning framework is architecture-agnostic and generalizes effectively beyond CLIP.
>
> **(Continued in next message)**

---

> ### Author Response · Authors · 2025-11-21
> **Response to Reviewer WPAs (part 2).**
>
> > W3&Q4: Self-distillation bias. HS uses fine-grained layer as the teacher. When the finest layer mis-aligns, the error may propagate. Have the authors explored soft teacher mixing (e.g., weighted fine + medium layers) or consistency regularization to mitigate this?
>
> We clarify that UPrompt is robust to self-distillation bias.
>
> On one hand, the finest-grained teacher signals are detached during optimization (L239-L240). This means gradients from coarse-level training do not flow back to the finest-grained layer, preventing any misalignment errors in the teacher from propagating downward. As stated in Prop. 2, this ensures gradient isolation. Thus, even if the finest layer misaligns temporarily, it does not corrupt the coarse-level representations. On the other hand, UPrompt's bidirectional design provides top-down context complements by refining fine-grained features using global information.
>
> This synergy ensures that even if the finest layer encounters local misalignments, CE supplies corrective context from coarser levels, creating a self-correcting loop. Table 5 demonstrates this: HS substantially improves coarse-layer performance (I2T R@1: 75.4%→82.8%). If fine layers frequently mis-aligned, we would observe performance degradation rather than consistent improvement.
>
> We conducted soft teacher mixing experiments on Flickr30K as suggested, with learnable weights (summing to 1) for fine and medium layers:
>
> |Teacher Strategy|I2T R@1|I2T R@5|T2I R@1|T2I R@5|
> |-|-|-|-|-|
> |Fine only (Ours)|93.8|99.4|83.6|96.3|
> |Fine + Medium (soft mixing)|93.7|99.2|83.9|96.3|
>
> The marginal difference confirms fine-layer supervision is robust. HS itself functions as consistency regularization by constraining KL divergence between coarse and fine distributions (Eq. 9), enforcing semantic alignment across granularities.
>
> > W4: Complexity analysis could be deeper: CE’s cross-granularity attention likely adds overhead, but FLOPs/memory/throughput statistics are not that detailed. Fig.5 is very complex and a little bit hard to understand.
>
> The computational overhead introduced by the cross-granularity attention in the Coarse-to-Fine Cascaded Enhancement (CE) is limited. As shown in Fig. 5, the variation in computational cost across different granularity settings is primarily driven by the number of visual tokens rather than the attention operation itself. More importantly, our UPrompt achieves comparable performance to the PSRC baseline while requiring only 1/3 of its computational cost. The effectiveness of the CE operation, which requires minimal computational overhead, is evident: according to Table 5, CE improves the Image-to-Text R@1 on Flickr30K from 92.7% to 93.8%.
>
> We have simplified Fig. 5 by averaging HM across OxfordPets and Caltech101 and added performance trend lines to better illustrate the efficiency-accuracy trade-offs. The revised figure will be included in the updated manuscript.
>
> > W5: Statistical significance: averages are reported across datasets, but per-dataset variance or confidence intervals are missing. Also, Table 6 in supp has wrong bold for UCF task. Is this a typo? This increases my skepticism towards the overall soundness of the reported results.
>
> We thank the reviewer for identifying the typographical error in Table 6, which will be corrected. Regarding statistical significance, we will provide error bar analysis in the final manuscript to ensure robust statistical evaluation across all reported results.
>
> > Q1: Experiments fix the input at 224×224 (14×14 tokens). If higher-resolution inputs (336/384) or denser backbones (ViT-L/H) are used, does the optimal number of granularities K change? Could you provide a resolution × layer-count ablation to disambiguate “benefit from more layers” vs. “compensation for limited resolution” or at least ablation study results with another resolution?
>
> We provide the resolution × layer-count ablation on Flickr30K (I2T R@1). Starting from the coarsest level and progressively adding finer levels, layer-count increases from 1 to 3:
>
> |Resolution|layer-count 1|layer-count 2|layer-count 3|
> |-|-|-|-|
> |224×224|82.8|89.4|93.8|
> |336×336|83.3|92.2|95.1|
>
> This ablation disambiguates "benefit from more layers" vs. "compensation for limited resolution." At 336×336 with 1.5× more tokens, multi-granularity remains the dominant factor for performance improvement. Increasing resolution alone (layer-count 1: 82.8→83.3) provides minimal gain, while multi-granularity (layer-count 1→3) drives substantial improvements at both resolutions. The optimal layer-count 3 is stable across resolutions, confirming our method addresses semantic hierarchy independent of spatial token density.
>
> Regarding denser backbones  (ViT-L/H) , Table 8 in Appendix B.3 shows UPrompt with ViT-L consistently outperforms other methods using the ViT-L backbone across MSCOCO and Flickr30K. The consistent improvements confirm multi-granularity benefits persist with denser backbones.
>
> **(Continued in next message)**

---

> > ### Author Response · Authors · 2025-11-21
> > **Response to Reviewer WPAs (part 3).**
> >
> > > Q5: Currently the inference layer is manually chosen. Could a lightweight gating or uncertainty-based controller dynamically select the appropriate granularity at test time for adaptive efficiency?
> >
> > We thank the reviewer for this thoughtful suggestion on improving adaptive efficiency. Dynamic granularity selection is orthogonal to our contribution and can be readily integrated with existing uncertainty-based methods. Our framework provides granularity-specific representations with semantic consistency via hierarchical supervision, enabling any uncertainty estimator (e.g., prediction entropy, feature norm) to select the appropriate level at test time without additional training. This modular design allows practitioners to combine our method with established adaptive inference techniques when adaptive efficiency is prioritized over explicit control of the performance-cost trade-off.
> >
> > [1] Wah et al. "The caltech-ucsd birds-200-2011 dataset." 2011.
> >
> > [2] Xian et al. "Zero-shot learning—a comprehensive evaluation of the good, the bad and the ugly." TPAMI 2018.

---

> > > ### Comment · Reviewer_WPAs · 2025-11-26
> > >
> > > Thanks the authors for the detailed responses. The responses address many of my concerns. I will keep the score to be the same.

---

> > > > ### Author Response · Authors · 2025-11-27
> > > > **Thank you.**
> > > >
> > > > We thank Reviewer WPAs for their constructive feedback and for recognizing that our responses have addressed their concerns. The suggested experiments have significantly strengthened our manuscript, and we remain happy to provide any further clarifications if needed.

---

### Author Response · Authors · 2025-11-27
**Global Response and Revised Manuscript Upload**

We thank all reviewers for their thoughtful and constructive feedback aimed at improving the quality of our work. The reviewers agree that the paper is *well-organized and easy to follow* (**WPAs**, **GiLu**), *addresses a clear and important gap* in single-granularity VLM prompt learning (**gXNe**, **WPAs**), and provides *comprehensive and rigorous experiments* with extensive ablations (**WPAs**, **gXNe**, **KDVr**). Reviewers also appreciate our *innovative U-Net-inspired bidirectional framework* (**gXNe**, **GiLu**), *sound theoretical proofs* (**WPAs**, **gXNe**), and *practical flexibility* for performance-efficiency trade-offs (**gXNe**, **WPAs**).

We have uploaded a revised version of the manuscript improving the clarity and including the changes suggested by the reviewers. **We highlight in blue all the changes for clearer visualization.** Please note that the paper lines and table/figure numbers we mentioned in the responses to the reviewers refer to the original paper and not the revised one. Below we provide a summary of the main additions to the revised manuscript.

## Novelty Clarification

**Limited novelty concerns (GiLu)**: We strengthened **L157-159** to emphasize the fundamental differences between UPrompt and existing multi-level methods (TAP, HiCroPL), clarifying that our U-shaped architecture establishes parallel granularity pathways with explicit bidirectional information flow rather than treating granularities as independent modules.

## Enhanced Clarity and Technical Details

**Unclear method description and Figure 2 (KDVr)**: We revised **Figure 2** with clearer legends and flow annotations, especially for the visual pathway construction.

**On prompt integration (KDVr)**: We added explicit explanation in **L177-L183** clarifying how multi-granularity visual patch tokens are constructed through progressive pooling and how they are combined with learnable prompts before being fed into the vision encoder.

**How the prompts are gradually made (KDVr)**: We added clarification in **L191-193** that multi-granularity text construction leverages LLMs, with reference to implementation details in Sec. 4.

**Self-distillation bias concerns (WPAs, gXNe)**: We added explanation in **L246-250** clarifying that detaching the finest-level teacher prevents error propagation from coarse-level training, while CE enables self-correction across granularities.

## Additional Experiments

**Complexity analysis clarity (WPAs)**: We simplified **Figure 5** by averaging HM across datasets and adding performance trend lines for better readability.

**Granularity interval strategy validation (GiLu)**: We added **Table 5** comparing different downsampling strategies (14×14→7×7→4×4 vs. 14×14→5×5→2×2) to validate our U-Net-inspired design choices.

**LLM dependency and reproducibility (WPAs, GiLu, gXNe, KDVr)**: We added **Table 6** evaluating text hierarchy generation across Qwen3-4B, Qwen3-14B, and Llama3-8B, demonstrating stable performance and addressing reproducibility concerns.

**Resolution vs. multi-granularity benefits (WPAs, KDVr)**: We added **Table 8** analyzing performance at 224×224 vs. 336×336 resolutions across different granularity levels, disambiguating multi-granularity benefits from resolution improvements.

**Statistical significance (WPAs)**: We added **Table 10 in Appendix B.3** with error bar analysis on cross-dataset evaluation across three independent runs, demonstrating remarkable stability.

**Generalization to other VLM architectures (WPAs)**: We added **Table 13 in Appendix B.5** with experiments on alternative VLM backbones (SigLIP, EVA-CLIP), validating architecture-agnostic generalization.

**Rule-based text hierarchy validation (WPAs)**: We added **Table 14 in Appendix B.6** with experiments on CUB-200 and AWA2 using purely rule-based attribute hierarchies, confirming our method works effectively without LLM-generated priors.

**Fine-layer supervision reliability (WPAs, gXNe)**: We added **Table 15 in Appendix B.7** comparing single fine-layer supervision against mixed fine+medium-layer supervision, validating the sufficiency of fine-grained teacher signals.

We again thank the reviewers for their thoughtful and insightful feedback. We are happy to address any further questions or provide additional clarifications.

---

### Author Response · Authors · 2025-12-03
**Summary for Area Chair and Senior Area Chair**

Dear Area Chair and Senior Area Chair,

We thank you for your time and the reviewers for their constructive feedback. Reviewers recognize the paper is *well-organized* (**GiLu, WPAs**) and addresses an *important gap* (**gXNe, WPAs**) through an *innovative and interesting design* (**gXNe, KDVr**) with *comprehensive experiments* (**WPAs, gXNe**). **WPAs** stated "*lean toward accept*" and **KDVr** stated "*leaning towards accept*" in their reviews.

We have uploaded a revised manuscript with all suggested improvements highlighted in blue. Below, we summarize our responses to the primary concerns:

> (1) Novelty, e.g., Differences from TAP and HiCroPL (**GiLu**)

We added clarifications distinguishing our parallel bidirectional pathway design from the static or independent modules used in TAP and HiCroPL, highlighting our unique cascaded enhancement and hierarchical supervision mechanisms.

> (2) Robustness to Different LLMs and Effectiveness of Rule-based Hierarchies (**WPAs, GiLu, gXNe, KDVr**)

We conducted additional experiments using Qwen3-4B and Qwen3-14B (Table 6), and implemented rule-based hierarchies without LLMs (Appendix B.6, Table 14). Results confirm our method is robust and effective regardless of the specific text generation source.

> (3) Impact of potential misalignment in fine-grained teacher (**WPAs, gXNe**)

We clarified the gradient isolation mechanism and highlighted results in Table 5 demonstrating that supervision improves coarse-level performance, confirming a self-correcting effect. Additional "soft teacher" experiments further validated robustness.

> (4) Experiments with different resolutions (**WPAs, KDVr**)

We added ablation studies (Table 8) comparing performance at different resolutions (224 vs. 336) to explicitly decouple the benefits of multi-granularity modeling from resolution scaling.

> (5) Dynamic granularity selection (**WPAs, GiLu**) and different interval strategies (**GiLu**)

We discussed the framework's compatibility with dynamic selection mechanisms and validated the method's effectiveness across different downsampling interval strategies in Table 5.

> (6) Clarification of method implementation and overview figure (**KDVr**)

We revised Figure 2 with detailed legends and flow annotations to improve visualization, and updated Section 3.2 to explicitly clarify the construction process, distinguishing between visual token pooling and learnable prompts.

**Current Status:**

Reviewer **WPAs** provided a comprehensive review that covered key concerns shared by others. They actively engaged in the discussion and confirmed that our responses resolved their issues.

Reviewers **GiLu**, **gXNe**, and **KDVr** did not participate in the discussion phase. We have provided point-by-point responses and new experiments to answer their questions.

We sincerely thank you and the reviewers for the time and effort dedicated to this submission. We hope this summary assists in your final assessment.

---

### Meta-Review · Area_Chair_h1yw · 2026-01-02

**Summary:**

This paper proposes UPrompt, a U-Net-inspired bidirectional multi-granularity prompt learning framework for vision–language model (VLM) adaptation. This paper gets 6, 4, 4, 4 in first round. The main concerns are limited novelty, limited experiment setting, poor writting and representation. A rebuttal is provided to partially address these concerns. But this paper needs major revision to meet the standard of ICLR.

**Reviewer Concerns:**

Concerns of Reviewer WPAs are partially addressed.
Concerns of Reviewer GiLu are not fully addressed.
Concerns of Reviewer gXNe are not fully addressed.
Concerns of Reviewer KDVr are partially addressed.

**Reviewer Scores:**

Reviewer WPAs would not change their score.
Reviewer GiLu would not change their score.
Reviewer gXNe would not change their score.
Reviewer KDVr may change their score to 6.

---

### Decision · Program_Chairs · 2026-01-26

Reject